



# New photolytic converter for improving aircraft measurements of NO$_2$ via chemiluminescence.

Clara M. Nussbaumer[1], Uwe Parchatka[1], Ivan Tadic[1], Birger Bohn[2], Daniel Marno[1], Monica Martinez[1], Roland Rohloff[1], Hartwig Harder[1], Flora Kluge[3], Klaus Pfeilsticker[3], Florian Obersteiner[4], Martin Zöger[5], Raphael Doerich[1], John N. Crowley[1], Jos Lelieveld[1,6], and Horst Fischer[1]

[1]Max Planck Institute for Chemistry, Department of Atmospheric Chemistry, 55128 Mainz, Germany
[2]Institute of Energy and Climate Research, IEK-8: Troposphere, Forschungszentrum Jülich GmbH, 52428 Jülich, Germany
[3]Institute of Environmental Physics, Heidelberg Univsersity, 69120 Heidelberg, Germany
[4]Karlsruhe Institute of Technology, 76021 Karlsruhe, Germany
[5]Flight Experiments, German Aerospace Center (DLR), 82234 Oberpfaffenhofen, Germany
[6]Climate and Atmosphere Research Center, The Cyprus Institute, Nicosia, Cyprus

**Correspondence:** Clara M. Nussbaumer (clara.nussbaumer@mpic.de)

**Abstract.** Nitrogen oxides ($NO_x \equiv NO + NO_2$) are centrally involved in the photochemical processes taking place in the earth's atmosphere. Measurements of $NO_2$, particularly in remote areas where concentrations are of the order of pptv (parts per trillion by volume), are still a challenge and subject to extensive research. In this study, we present $NO_2$ measurements via photolysis-chemiluminescence during the research aircraft campaign CAFE Africa (Chemistry of the Atmosphere - Field Experiment in

Africa) 2018 around Cabo Verde as well as the results of laboratory experiments to characterize the photolytic converter used. We identify a memory effect within the conventional photolytic converter associated with high NO concentrations and rapidly changing water vapor concentrations, accompanying changes in altitude during aircraft measurements, which is due to the porous structure of the converter material. We test and characterize an alternative photolytic converter made from quartz glass which improves the reliability of $NO_2$ measurements in laboratory and field studies.

## 1   Introduction

$NO_x$ (nitrogen oxides) represent the sum of NO (nitric oxide) and $NO_2$ (nitrogen dioxide) which can rapidly interconvert in the atmosphere in the presence of sunlight and $O_3$ (ozone) as shown in Reactions (R1) and (R2) (Jacob, 1999).

$$NO + O_3 \rightarrow NO_2 + O_2 \tag{R1}$$

$$NO_2 + h\nu \rightarrow NO + O(^3P) \qquad\qquad O(^3P) \xrightarrow{O_2} + O_3 \tag{R2}$$

Considering only these two reactions in atmospheric $NO_x$ chemistry, the so called Leighton ratio represents $NO_2$, NO and $O_3$ in photostationary state (PSS) as shown in Equation (1) (Leighton, 1961). $k_{NO+O_3}$ is the rate coefficient of Reaction (R1) and



$j_{NO_2}$ is the photolysis frequency for $NO_2$ in Reaction (R2).

$$\frac{k_{NO+O_3} \times [NO] \times [O_3]}{j_{NO_2} \times [NO_2]} = \phi \cong 1 \tag{1}$$

Several studies have shown that the Leighton ratio as presented in Equation (1) is only valid for highly polluted environments whereas in other regions, besides $O_3$, oxidized halogen species and peroxy radicals ($HO_2$ and $RO_2$) significantly contribute to the oxidation of NO to $NO_2$ and require an extension of the Leighton expression for a reliable calculation of PSS $NO_2$ concentrations as presented in Sect. 2.4. (Mannschreck et al., 2004; Griffin et al., 2007; Ma et al., 2017; Reed et al., 2016).

 $NO_x$ can be emitted from either natural and anthropogenic sources, with the latter dominating globally. Natural emissions
include for example biogenic soil emissions, biomass burning and lightning. Anthropogenic emissions are mainly from combustion processes in vehicles or from power and industrial plants which contribute almost two thirds to the global burden (Pusede et al., 2015; Ganzeveld et al., 2002; Logan, 1983). Nitrogen oxides are, together with volatile organic compounds, important precursors to tropospheric ozone which can be a hazard to plant, animal and human health, causing multiple diseases regarding the cardiovascular and respiratory system (Nussbaumer and Cohen, 2020; Nuvolone et al., 2018; Lippmann, 1989).
$NO_x$ additionally promote the formation of acid rain (through conversion to $HNO_3$) - hazardous to many ecosystems - and are a threat to human health themselves (Boningari and Smirniotis, 2016; Greaver et al., 2012). Beyond that, $NO_x$ control the abundance of OH radicals which regulate the oxidizing capacity of the atmosphere (Levy, 1971; Lelieveld and Dentener, 2000). Due to the health implications and the impact on atmospheric photochemical processes, it is highly relevant to measure and monitor ambient $NO_x$ concentrations with sophisticated instruments which provide reliable concentration measurements,
especially also in remote areas where NO and $NO_2$ are low. More specifically, this requires a low instrumental background which - particularly for $NO_2$ - is often impacted by unwanted chemical processes which can lead to artifact signals (Reed et al., 2016; Andersen et al., 2020; Jordan et al., 2020).

 Many different measurement techniques have been deployed to measure nitrogen oxides such as cavity enhanced absorption spectroscopy (and variants, e.g. cavity attenuated phase shift spectroscopy (Ge et al., 2013; Kebabian et al., 2005), cavity ring
down spectroscopy (O'Keefe and Deacon, 1988) and others (Zheng et al., 2018)), differential optical absorption spectroscopy (Hüneke et al., 2017; Winer and Biermann, 1994) and laser induced fluorescence (Thornton et al., 2000; Javed et al., 2019) for $NO_2$ or absorption spectroscopy for NO (Ventrillard et al., 2017). However, detection of NO and $NO_2$ via chemiluminescence (CLD) is likely the most common technique for the measurement of nitrogen oxides in the atmosphere and is distinguished by the simultaneous in-situ measurement of both, NO and $NO_2$, low detection limits and the deployability in research aircrafts
at high altitudes for measurements in the upper troposphere (Pollack et al., 2010; Reed et al., 2016; Tadic et al., 2020). The measurement principle is based on the reaction of nitric oxide and ozone which yields electronically excited $NO_2$ ($NO_2^*$) which (along with physical quenching) returns to the electronic ground state by fluorescence whereby a photon of a wavelength $>600\,nm$ is emitted, which can be detected by a photomultiplier tube. The resulting signal is proportional to the initial NO concentration (Clough and Thrush, 1967). For nitrogen dioxide detection, $NO_2$ is first converted to NO. The standard method
for this conversion is the use of a catalytic converter, in which $NO_2$ passes through a heated molybdenum converter where it is reduced by Mo to NO ($Mo + 3\,NO_2 \rightarrow MoO_3 + 3\,NO$). However, high temperatures ($300\text{ - }350\,°C$) in the converter along with





catalytic surface effects lead to interferences with other atmospheric compounds that can be converted to $NO_2$ such as HONO (nitrous acid), $HNO_3$ (nitric acid) or PAN (peroxyacyl nitrate) and bias the measurement (Demerjian, 2000; Villena et al., 2012; Jung et al., 2017). An alternative and widespread method is the use of a photolytic converter (photolysis-chemiluminescence:
P-CL), also referred to as blue light converter, which utilizes LEDs emitting at a wavelength of around 395 nm to dissociate $NO_2$ to NO (Pollack et al., 2010; Reed et al., 2016; Tadic et al., 2020; Ryerson et al., 2000). Interferences (as described above) in the blue light converter are still possible, but to a significantly lesser extent. Reed et al. (2016) investigated potential interferences in a photolytic converter which are related to the presence of PAN, methyl peroxy nitrate (MPN, $CH_3O_2NO_2$) or pernitric acid (PNA, $HO_2NO_2$). These compounds are $NO_2$ reservoir species and their decomposition (to $NO_2$) is dependent
on the temperature, the pressure and the residence time in the blue light converter (Nault et al., 2015; Fischer et al., 2014). Please note, that none of these compounds are photolyzed in the blue light converter and only subject to thermal decomposition (Reed et al., 2016; Tadic et al., 2020). Generally, increasing temperature and residence time promote the decay of thermally unstable trace gases and the release of $NO_2$ which is further described in Section 2.5 (Reed et al., 2016). The CLD detects a signal (which we call $NO_c$ signal) which is composed of the ambient NO concentration and the ambient $NO_2$ concentration
multiplied by the conversion efficiency $C_e$ according to Equation (2).

$$[NO_c] = [NO] + C_e \times [NO_2] \tag{2}$$

The conversion efficiency describes the fraction of $NO_2$ that is converted to NO in the converter and can be thought of as the NO yield from $NO_2$. Its value is dependent on the optical output of the LEDs as well as the $NO_2$ residence time and the pressure in the converter. $C_e$ is therefore in competition with unwanted formation of $NO_2$ from $NO_2$ reservoir species. For example, a
longer residence time increases the conversion efficiency, but could potentially increase the amount of $NO_2$ reservoir species that decay in the converter, which takes place according to first order kinetics which is described in more detail in Section 2.5. The $NO_2$ concentration is calculated from the difference in the signal with and without use of the photolytic converter: $[NO_2] = ([NO_c]-[NO])/C_e$ (Sadanaga et al., 2010; Tadic et al., 2020; Ryerson et al., 2000).

While NO measurements are generally reliable and well-understood, $NO_2$ measurement techniques utilizing the conversion
of $NO_2$ to NO are subject to extensive research. Hosaynali Beygi et al. (2011) found a strong deviation from the Leighton ratio at low $NO_x$ concentrations between 5 and 25 pptv despite the inclusion of $HO_2$, $RO_2$ and halogen oxides suggesting the occurrence of a so far unknown atmospheric oxidant. Frey et al. (2015) also reported higher measured $NO_2$/NO ratios than expected from PSS based on measurements in Antarctica and hypothesized the presence of an additional oxidant or a measurement bias. This is in line with findings and suggestions by Silvern et al. (2018) based on observations during the aircraft
campaign SEAC[4]RS over the United States of America. Reed et al. (2016) examined the described deviation through the laboratory investigation of potential $NO_2$ interferences of thermally unstable trace gases such as peroxyacyl nitrate (PAN) within the photolytic converter in comparison to laser-induced fluorescent $NO_2$ measurement and found that this could contribute to the higher than expected $NO_2$ concentrations measured by P-CL instruments. Jordan et al. (2020) investigated interferences in a photolytic converter made from quartz glass and showed how the converter conditions affect the conversion efficiency and the



artifact signal (caused by $NO_2$ reservoir species). The correct adjustment of the conditions, preferably including low pressure, high flow rates and small temperature variations, can minimize interferences. Andersen et al. (2020) reported the measurement of a significant $NO_2$ measurement bias during ground-based observations in the remote marine tropical troposphere with a conventional blue light converter which was related to its porous walls. They were able to eliminate this effect by implementation of a photolytic converter made from quartz glass which reduced the overall measurement uncertainty by around 50 %.

An additional challenge is the significant decrease in the $NO_2$/NO ratio with altitude. At the surface at daytime, $NO_2$ concentrations are approximately two to four times higher than NO concentrations. The $NO_2$/NO ratio decreases by around one order of magnitude when going from the lower to the upper troposphere which increases the uncertainty when deriving $NO_2$ mixing ratios using Equation (2) (Travis et al., 2016; Silvern et al., 2018; Logan et al., 1981). At the same time, the concentration of $NO_2$ reservoir species such as PNA or MPN is significantly higher in the upper troposphere compared to that at the surface and

consequently interferences are more likely to occur at high altitudes (Nault et al., 2015; Kim et al., 2007). These aspects result in particularly strict requirements regarding airborne $NO_2$ measurements.

    In this study, we describe a modified blue light converter (BLC) originally purchased from Droplet Measurement Technologies, which we have deployed in $NO_2$ measurements via photolysis-chemiluminescence during the research aircraft campaign CAFE Africa (Chemistry of the Atmosphere: Field Experiment in Africa) and also in laboratory investigations. We show how

high NO concentrations and rapidly changing water vapor concentrations affect the instrumental background and induce a memory effect which cannot be corrected retrospectively. This is particularly relevant to aircraft measurements where water vapor concentrations are subject to rapid changes due to variations in flight altitude, but also to all other application areas. The photolytic converter and similar designs are widely used for field measurements of $NO_2$ all across the world (e.g. Andersen et al. (2020); Jung et al. (2017); Xu et al. (2013); Breuninger et al. (2013); Fuchs et al. (2010); Reidmiller et al. (2010); Crowley

et al. (2010); Sather et al. (2006)) and can provide reliable results for stationary use and locations with only little variations in ambient NO and low humidity levels, but suffer from enhanced uncertainty in other applications. We propose the elimination of any direct contact points between the sample gas and the porous inner converter surface and have developed an alternative photolytic converter entirely made from quartz glass. Highly reflective properties are achieved by an outer mantel made from optical PTFE (polytetrafluoroethylene, also known as teflon). The new converter shows promising results in the laboratory

regarding its application in field studies for more reliable $NO_2$ measurements. We do not claim to be the first to present an alternative quartz glass converter for P-CL measurement of $NO_2$. However, we are first to point out the technical difficulties in the application of conventional $NO_2$ converters in airborne studies and believe the presented results to be a guidepost for future $NO_2$ aircraft measurements via photolysis-chemiluminescence.

## 2   Observations and methods

### 2.1   Instrument

All $NO_x$ measurements were performed using a modified commercially available two-channel chemiluminescence instrument (detector from ECO Physics CLD 790 SR, Dürnten, Switzerland) as described by Tadic et al. (2020) operated at a total gas flow





of 3 SLM, equally divided into the two channels. NO concentrations are measured in the first channel, also referred to as the NO channel, through formation of $NO_2^*$ via reaction with $O_3$. The resulting excited $NO_2^*$ emits a photon ($> 600\,nm$) detected

by a photomultiplier tube, preamplifier set up and recorded as counts per second. The second channel, also referred to as $NO_c$ channel, is structurally identical except for the implementation of a photolytic converter which converts a known fraction of $NO_2$ to NO prior to the reaction with $O_3$ and is operated at a constant pressure of $110\,hPa$ ($105\,hPa$ during the CAFE Africa field experiment). $NO_2$ concentrations are obtained from the difference in counts from each channel and the conversion efficiency $C_e$ as described above (see Equation 2). We use a blue light converter purchased from Droplet Measurement Technologies

equipped with UV-LEDs emitting at a wavelength of $398\,nm$ (FWHM = $16\,nm$) which is shown in Figure 1a (Tadic et al., 2020). Originally, the inner material is made of porous, optically active PTFE (polytetrafluoroethylene) for providing highly reflective properties. To reduce surface effects the converter was equipped with a quartz cylinder covering approximately half of the PTFE surface (the gas still gets in touch with the PTFE surface in the ring channel and through the head piece). The sample gas enters the converter sideways into the ring channel and reaches the inner tube via the PTFE head piece which has

four circular recesses, one for each UV LED. The sample gas outlet proceeds analogously. The inner volume of the converter is $V = 78\,cm^3$ which gives a residence time of $t = \dfrac{V \times 60\,s\,min^{-1}}{F} \times \dfrac{p}{p_{standard}} = \dfrac{78\,cm^3 \times 60\,s\,min^{-1}}{1500\,cm^3\,min^{-1}} \times \dfrac{110\,hPa}{1013\,hPa} = 0.34\,s$. The conversion efficiency for this photolytic converter operated under the conditions described above is approximately $20\,\%$ which was determined via gas phase titration (GPT) of NO with ozone. The results obtained with the described blue light converter were compared to a newly-developed photolytic converter completely made from quartz glass which is shown in

Figure 1b. For maintaining the reflective properties of the blue light converter, the new quartz glass converter was jacketed with optical PTFE which provides diffuse reflectance of $> 99\,\%$ in the wavelength range $350$ - $1500\,nm$ (SphereOptics GmbH, 2017). The volume of the new converter is $77\,cm^3$ which gives a residence time of $t = 0.33\,s$ and a conversion efficiency of approximately $14\,\%$ under the operating conditions. The main difference between the two converters is that the sample gas flow does not get in direct contact with the porous surface of the material for the new quartz glass converter. The reaction chambers

(where the reaction of NO and $O_3$ takes places) are operated at a constant temperature of $25\,°C$ and a pressure of $7$ - $8\,mbar$ in order to minimize quenching of $NO_2^*$ by other molecules. The dry ozone flow is humidified with water vapor for maintaining a constant humidity level at all times.

Besides the photons emitted from relaxation of $NO_2^*$, the PMT signal also includes detected photons from interference reactions, for example the reaction of $O_3$ with alkenes (Alam et al., 2020), as well as a dark current signal. Therefore, a

pre-chamber measurement is operated for 20 seconds every 5 minutes where ozone is added to the sample gas flow. The residence time in the pre-chamber allows for the reaction of $O_3$ and NO and the relaxation of $NO_2^*$ before entering the main reaction chamber. It is not long enough to convert interfering compounds which then occurs in the following main chamber. Consequently during pre-chamber measurements, the PMT signal only includes the interfering signal and the dark current signal (Ridley and Howlett, 1974; ECO PHYSICS AG, 2002). We subtracted the interpolated signal obtained during pre-

chamber measurements from the signal detected during main chamber measurements in order to obtain the signal generated from NO.





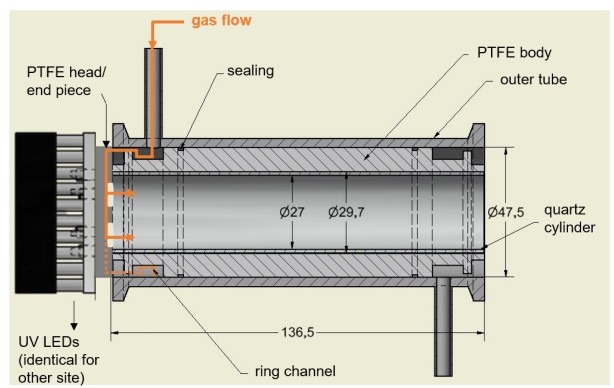

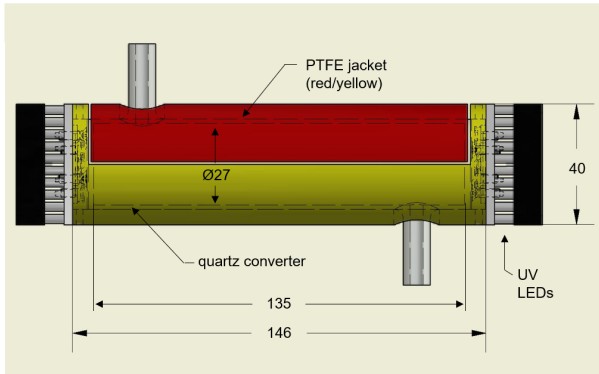

(a) conventional blue light converter with quartz glass cylinder

(b) new photolytic quartz glass converter

**Figure 1.** Sketches of the photolytic converters applied in this study.

The instrumental background of each channel is determined via zero (synthetic) air measurements from a gas cylinder and can be converted to mixing ratios using calibration measurements with a known NO concentration which defines the sensitivity (counts s$^{-1}$ per ppbv (parts per billion by volume) of each channel towards NO as shown in Eq. (3) (after pre-chamber corrections). The signal detected from zero air measurement (counts(zero air)) is subtracted from the signal detected from NO calibration (counts(NO calibration)) and divided by the absolute concentration of the NO calibration (c(NO calibration)) to calculate the sensitivity. Dividing the signal detected from zero air measurements by this value gives the background concentration in mixing ratios, e.g. ppbv.

$$\text{sensitivity} = \frac{\text{counts(NO calibration)} - \text{counts(zero air)}}{\text{c(NO calibration)}} \qquad \text{c(background)} = \frac{\text{counts(zero air)}}{\text{sensitivity}} \qquad (3)$$

Please note that the utilized zero air can include a trace concentration of $NO_x$. The manufacturer specifies the maximum concentration of $NO_x$ to be 0.1 ppmv (parts per million by volume) (Westfalen Gas Schweiz GmbH).

## 2.2 CAFE Africa field experiment

The CAFE Africa research campaign took place in August and September 2018 and included fourteen measurement flights (MF03 - MF16) which were performed with the HALO (High Altitude Long Range) research aircraft starting from the campaign base in Sal on Cabo Verde (16.75 ° N, 22.95 ° W). We included data measured during the measurement flights MF10, MF12, MF13, MF14 and MF15 in this analysis (MF11 was a nighttime flight and therefore excluded) for which CLD $NO_2$ measurements were available. An overview of the flight tracks is presented in Figure 2. More details on the campaign can be found in Tadic et al. (2021).

NO and $NO_2$ were measured via photolysis-chemiluminescence with the instrument described in Sect. 2.1 using the conventional blue light converter equipped with the quartz glass cylinder, operated at a temperature of 313 K and a pressure of





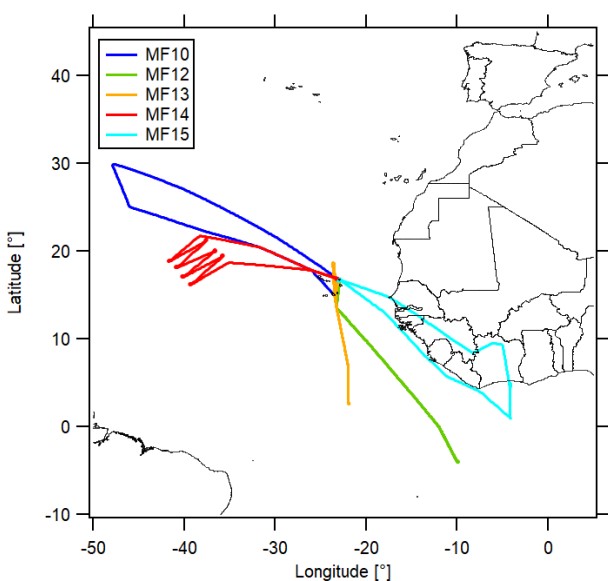

**Figure 2.** Geographic overview of the measurement flight tracks included in this analysis.

105 hPa (0.32 s residence time). Please note that it was not possible to measure the temperature inside the converter. Instead, the temperature of the gas outflow from the converter in the ring channel was measured which we assume to be identical to the inner temperature. Zero air measurements and NO calibrations were performed regularly to determine the variability in the instrumental background and the sensitivity of the channels. The ambient measurement was interrupted every 1 - 2 h by one

minute zero air measurement, followed by one minute NO calibration and another one minute zero air measurement. These calibration-background-cycles (CB-cycles) were performed 4 - 6 times during each measurement flight. We linearly interpolated these background and calibration measurements to each entire flight. The NO data were processed as described by Tadic et al. (2021) (5 pptv detection limit and 6 % relative uncertainty). Please note that the instrumental background for the NO data was determined by nighttime measurements of NO instead of zero air measurements as presented by Tadic et al. (2021) and

previously described by Lee et al. (2009). For the $NO_2$ data, the $NO_c$ channel sensitivity and the background concentration were calculated after pre-chamber correction according to Equation (3). The ambient $NO_c$ signal was divided by the channel sensitivity, accordingly. Final $NO_2$ concentrations were determined via Equation (4) which includes subtraction of the calculated (and interpolated) $NO_c$ background concentration and the NO concentration in the NO channel, and dividing by the conversion efficiency of the blue light converter which was $24.2 \pm 2.8$ % during the campaign.

$$c(NO_2) = \frac{c(NO_c) - c(background_{NO_c}) - c(NO)}{C_e} \qquad (4)$$



## 2.3 Further measurements

Additional measurements of atmospheric trace gases during CAFE Africa including $O_3$, CO, $CH_4$, $HO_2$, OH, $NO_2$ and water vapor as well as the photolysis frequencies $j_{NO_2}$ and $j_{PNA}$ were used in this study. $O_3$ was measured via UV absorption and chemiluminescence with the FAIRO (Fast AIRborne Ozone) instrument (total measurement uncertainty of 2.5 %, Zahn et al. (2012). CO and $CH_4$ were measured via quantum cascade laser absorption spectroscopy (total measurement uncertainty of 4.3 % and 0.3 %, respectively, Schiller et al. (2008)). $HO_2$ and OH were measured with the custom-built HORUS (HydrOxyl Radical measurement Unit based on fluorescence Spectroscopy) instrument via fluorescence spectroscopy (Novelli et al., 2014; Marno et al., 2020). Please note that these data are still preliminary and the measurement uncertainty is estimated at 50 %. Additional $NO_2$ concentrations for comparison were measured via differential optical absorption spectroscopy (miniDOAS) with a detection limit of about 5 pptv and an uncertainty depending on the altitude and cloud cover of typically 40 pptv (Hüneke et al., 2017; Kluge et al., 2020). Water vapor was measured via direct absorption by the tunable diode laser system SHARC (Sophisticated Hygrometer for Atmospheric ResearCh) (accuracy of 5 %, detection limit typically in the range of 2 - 3 ppmv) (Kaufmann et al., 2018). The photolysis frequencies were calculated from actinic flux densities measured with a spectral radiometer (Meteorologie Consult GmbH, Metcon, Koenigstein, Germany) (uncertainty < 15 %) (Bohn and Lohse, 2017). Please note that all measurement data were converted to a uniform timescale with a 1 min time resolution as a basis for this analysis.

## 2.4 $NO_2$ calculations

For calculating the photostationary state $NO_2$ concentrations during CAFE Africa, we assume that $NO_2$ production occurs through reaction of NO with $O_3$ (Reaction (R1)), $HO_2$ (Reaction (R3)) and $RO_2$ (Reaction (R4)). Tadic et al. (2021) showed that $RO_2$ is well represented by $CH_3O_2$ during CAFE Africa via model simulations (80 % at 200 hPa altitude and up to 90 % below) which we therefore use as surrogate for describing all organic peroxy radicals. In analogy to Leighton (1961), we describe $NO_2$ loss by photo dissociation as shown in Reaction (R2). Other loss pathways for $NO_2$ for example via OH can be neglected (< 1 %) (Bozem et al., 2017).

$$NO + HO_2 \rightarrow NO_2 + OH \tag{R3}$$

$$NO + CH_3O_2 \rightarrow NO_2 + CH_3O \tag{R4}$$

$NO_2$ concentration in photostationary state can therefore be obtained via Equation (5) whereas the concentration of $CH_3O_2$ is calculated by help of Equation (6) which was derived by Bozem et al. (2017). For the calculation via Equation (6) we assume that $CH_3O_2$ and $HO_2$ formation occur through $CH_4$ and CO oxidation, respectively. We estimate an uncertainty of around 20 % resulting from these assumptions. Propagating the measurement uncertainties of $HO_2$, $CH_4$ and CO suggests a 50 % uncertainty





in the calculated $CH_3O_2$ data. The $NO_2$ PSS data have an uncertainty of 22 % regarding the trace gas measurements according to Gaussian error propagation (uncertainty of rate coefficients is considered negligible).

$$[NO_2]^{PSS} = \frac{[NO] \times (k_{NO+O_3} \times [O_3] + k_{NO+HO_2} \times [HO_2] + k_{NO+CH_3O_2} \times [CH_3O_2])}{j_{NO_2}} \tag{5}$$

$$[CH_3O_2] = \frac{k_{CH_4+OH} \times [CH_4]}{k_{CO+OH} \times [CO]} \times [HO_2] \tag{6}$$

The temperature dependent rate coefficients were obtained from the data sheets of the IUPAC Task Group on Atmospheric Chemical Kinetic Data Evaluation (2021) (Atkinson et al., 2004, 2006).

### 2.5 Calculation of NO$_2$ reservoir species

We consider the $NO_2$ reservoir species PAN, MPN and PNA. PAN was measured during CAFE Africa via chemical ionization
mass spectrometry (CIMS) (Phillips et al., 2013). MPN and PNA were not measured and instead estimated via photostationary state calculations as suggested by Murphy et al. (2004). PNA ($HO_2NO_2$) production occurs through reaction of $HO_2$ and $NO_2$ (R5) while PNA loss is described by Reactions (R6)-(R8) either through thermal decomposition, photolysis or reaction with OH (Veres et al., 2015; IUPAC Task Group on Atmospheric Chemical Kinetic Data Evaluation, 2021; Atkinson et al., 2004). PSS $HO_2NO_2$ concentrations can then be calculated via Equation (7). $k$ is the rate coefficient for each reaction and $j_{PNA}$ is
the photolysis frequency for Reaction (R7).

$$HO_2 + NO_2 + M \rightarrow HO_2NO_2 + M \tag{R5}$$

$$HO_2NO_2 + M \rightarrow HO_2 + NO_2 + M \tag{R6}$$

$$HO_2NO_2 + h\nu \rightarrow products \tag{R7}$$

$$HO_2NO_2 + OH \rightarrow products \tag{R8}$$

$$[HO_2NO_2]^{PSS} = \frac{k_5 \times [HO_2][NO_2]^{PSS}}{k_6 + j_{PNA} + k_8 \times [OH]} \tag{7}$$

MPN production and loss terms are in analogy to PNA as shown in Reactions (R9)-(R11) except for the reaction with OH which is negligible (Nault et al., 2015; Browne et al., 2011; Murphy et al., 2004; Bahta et al., 1982). The calculation in PSS is





performed via Equation (8). $k$ represents the rate coefficients and $j_{MPN}$ is the photolysis frequency for Reaction (R11). During CAFE Africa, only the photolysis frequency $j_{PNA}$ was evaluated because reliable molecular data for MPN were missing. As suggested by Murphy et al. (2004) we assume identical UV cross sections of MPN and PNA and therefore $j_{MPN}$ to be identical

with $j_{PNA}$.

$$CH_3O_2 + NO_2 + M \rightarrow CH_3O_2NO_2 + M \tag{R9}$$

$$CH_3O_2NO_2 + M \rightarrow CH_3O_2 + NO_2 + M \tag{R10}$$

$$CH_3O_2NO_2 + h\nu \rightarrow products \tag{R11}$$

$$[CH_3O_2NO_2]^{PSS} = \frac{k_9 \times [CH_3O_2][NO_2]^{PSS}}{k_{10} + j_{MPN}} \tag{8}$$

In the photolytic converter PNA, MPN and PAN can decompose to $NO_2$ depending on the temperature, the pressure and the residence time $t$ according to first order kinetics. The resulting $NO_2$ artifact is determined via Equation (9).

$$
\begin{aligned}
[NO_2]_{artifact} = &[HO_2NO_2]^{PSS} \times (1 - exp(-k_6 \times t)) \\
&+ [CH_3O_2NO_2]^{PSS} \times (1 - exp(-k_{10} \times t)) \\
&+ [CH_3COO_2NO_2] \times (1 - exp(-k_{CH_3COO_2NO_2+M} \times t))
\end{aligned} \tag{9}
$$

Gaussian error propagation gives an uncertainty of 55 % for the calculated PNA and MPN data. We use the residence time according to the volume and the flow rate in the photolytic converter as described earlier. The actual value could deviate from the calculated one due to unknown flow-dynamics and temperature gradients. Assuming 30 % uncertainty in the residence time gives an overall uncertainty of around 60 % in the $NO_2$ formed from PNA and MPN in the photolytic converter.

## 3   Results and Discussion

### 3.1   Aircraft measurements

#### 3.1.1   $NO_2$ reservoir species

Figure 3a shows the vertical concentration profiles of the $NO_2$ reservoir species MPN and PNA according to photostationary steady state calculations (Equations (8) and (7)) as well as PAN measurements during CAFE Africa. MPN concentrations were



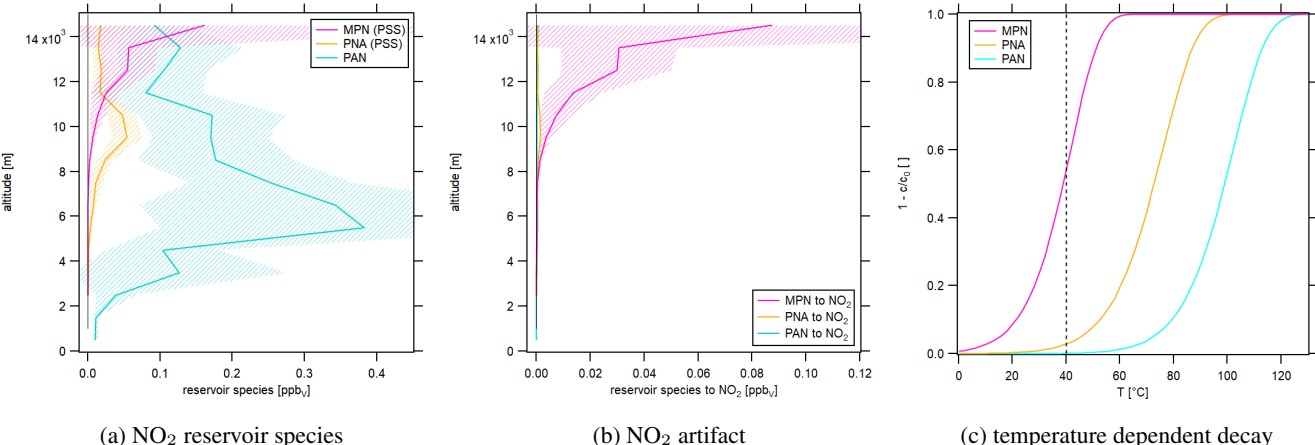

(a) NO$_2$ reservoir species        (b) NO$_2$ artifact        (c) temperature dependent decay

**Figure 3.** Vertical profiles of (a) NO$_2$ reservoir species MPN, PNA (from PSS calculations) and PAN (measured during CAFE Africa) and (b) NO$_2$ artifacts in the blue light converter from reservoir species according to first order decay. The shaded plot areas present the $1\sigma$ standard deviation resulting from averaging the concentrations at each considered altitude range. (c) Temperature dependent decay of the NO$_2$ reservoir species in the blue light converter.

close to zero at low altitudes up to 10 km and increased above, reaching $57 \pm 40$ pptv between 13 and 14 km altitude. The concentration increased further aloft but had a large variability. PNA mixing ratios were low below 8 and above 12 km altitude and showed peak concentrations of $54 \pm 21$ pptv between 9 and 10 km. PAN increased from ground level to mid-range altitudes with a maximum of $383 \pm 283$ pptv at 4 - 5 km. Concentrations subsequently decreased with altitude, reaching $92 \pm 44$ pptv at 14 - 15 km. Figure 3b shows the NO$_2$ artifact concentrations resulting from thermal decomposition of the reservoir species in

the blue light converter according to first order decay. It can be seen that only relevant artifact signals originated from MPN of which more than 50 % decomposed to NO$_2$ at the conditions present in the converter. 3 % of the ambient PNA was converted to NO$_2$. Even though atmospheric PAN concentrations, particularly at mid-range altitudes, were high, temperature, pressure and residence time in the blue light converter were too low for PAN to decay to NO$_2$. As an overview, Figure 3c shows the temperature-dependent decay ($1 - c/c_0$) of the discussed NO$_2$ reservoir species in the converter. The calculation is based on

constant pressure (105 hPa) and residence time (0.32 s). The temperature in the converter is shown with the black dashed line. Increasing temperature increases the decomposition share. It can be seen that, for PAN and PNA, the converter temperature would need to be significantly higher to observe a relevant decay (10 % decay of PNA at $> 50\,^\circ$C and of PAN at $\sim 80\,^\circ$C). In contrast for MPN, small changes in the temperature have a strong effect on the decomposing share (4 % per $^\circ$C at the steepest point). We show the time- and pressure-dependent decay of PAN, PNA and MPN in Figure S1 of the Supplement. PAN and

PNA decay only slightly depends on pressure at the given temperature and residence time. Please note that the residence time and the pressure are correlated, which we have neglected in this calculation. Based on these results, we recommend the implementation of a monitoring system for pressure and, more importantly, temperature within the photolytic converter which





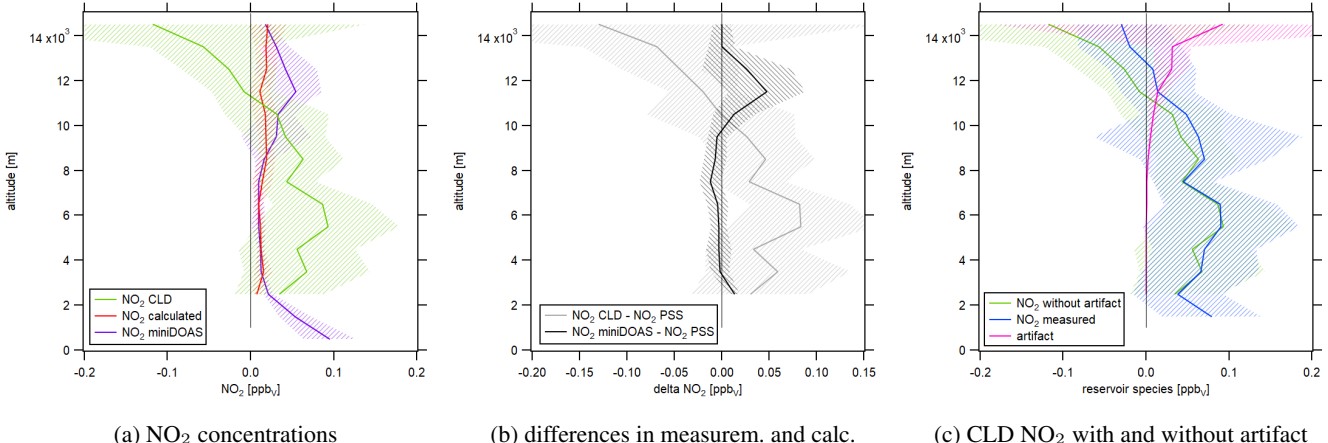

(a) NO$_2$ concentrations     (b) differences in measurem. and calc.     (c) CLD NO$_2$ with and without artifact

**Figure 4.** Vertical profiles of (a) NO$_2$ concentrations measured by the CLD, miniDOAS and calculated via Eq. (5), (b) the difference in NO$_2$ concentrations from PSS calculations and measurements and (c) CLD NO$_2$ concentrations with and without artifact. The shaded plot areas present the $1\,\sigma$ standard deviation resulting from averaging concentrations at each considered altitude range.

is difficult to implement in the commercially available blue light converter, but allows for a more accurate calculation of the decomposing share and consequently a reliable correction of the NO$_2$ signal.

We have subtracted the NO$_2$ artifact signal arising from the decomposition of MPN and PNA from the CLD NO$_2$ concentrations. Please note that the data coverage for the NO$_2$ artifact from MPN is 55 % and from PNA is 48 % (difference due to OH data coverage). We have interpolated the data used in the following sections to reach full coverage of the CLD NO$_2$ concentrations. Sometimes the data were incomplete at the start or the end of a measurement flight in which case we considered the averaged NO$_2$ artifact signal according to the vertical profile shown in Figure 3b as a function of the altitude.

### 290    3.1.2   Atmospheric NO$_2$ concentrations

Figure 4 shows the vertical profile of NO$_2$ concentrations measured via CLD in green, miniDOAS measurements in purple and NO$_2$ from PSS calculations in red. Calculated PSS NO$_2$ concentrations were on average $17 \pm 14$ pptv and approximately constant over the considered altitude range. At high altitudes, NO$_2$ from decomposing reservoir species exceeded the PSS values by around a factor of five. NO$_2$ concentrations measured by the miniDOAS instrument were $95 \pm 31$ ppt at ground level

and decreased with altitude up to 2 km. They were constant with $15 \pm 16$ pptv between 2 and 10 km altitude and agreed to within $\sim 85$ % to the calculated values. Concentrations increased again above reaching $54 \pm 31$ pptv between 11 and 12 km and decreased aloft with values similar to PSS NO$_2$ between 14 and 15 km. Average NO$_2$ concentrations measured by the CLD were $49 \pm 76$ pptv below 10 km altitude where decomposition of reservoir species did not play a role and decreased with altitude above. Figure 4b shows the calculated difference in NO$_2$ concentrations between PSS calculations and miniDOAS

measurements in black, and between PSS calculations and CLD measurements in gray. It is notable that NO$_2$ concentrations from PSS and miniDOAS measurements were nearly identical apart for a difference with a maximum value of $48 \pm 4$ pptv





between 10 and 13 km altitude. In contrast, CLD $NO_2$ concentrations were higher by $45 \pm 62$ pptv compared to the calculation up to 10 km altitude and lower at higher altitudes with a maximum deviation of more than 100 pptv between 14 and 15 km. Figure 4a shows that the $NO_2$ CLD mixing ratios are negative at high altitudes. This is an indicator of a wrongly measured background signal in the second channel. If the determined instrumental background was too high, Eq. (4) could return under-

estimated or even negative $NO_2$ concentrations. However, the CLD $NO_2$ data were not generally too small, but even enhanced at lower altitudes compared to PSS and miniDOAS data which may indicate the contribution of additional factors which we investigate in the following by the help of NO, $H_2O$ and $NO_2$ concentrations in the course of selected measurement flights. For comparison, Figure 4c shows the CLD $NO_2$ data with and without the calculated artifact signal. It can be seen that the data are

already negative before the subtraction of decomposing $NO_2$ reservoir species.

### 3.1.3 Influence of atmospheric water vapor

Atmospheric water vapor concentrations are highest at ground-level and decrease with increasing altitude. As an example, the vertical concentration profile of atmospheric water vapor during CAFE Africa is shown in Figure S2 of the Supplement. Accordingly, altitude changes during aircraft measurements introduce rapid changes in relative humidity to the instruments

on-board.

Figure 5 shows a time series of NO, water vapor, and calculated and measured $NO_2$ concentrations during the measurement flights MF10 (Figure 5a) and MF12 (Figure 5b). NO concentrations varied between 0.005 and 0.56 ppbv for MF10 and between 0.005 and 0.46 ppbv for MF12. We have recently shown that enhanced NO concentrations in the morning and afternoon of MF12 were due to local, recent lightning activity (Nussbaumer et al., 2021). For MF10, enhanced NO concentrations at

high altitudes had their source over the African continent. At low altitudes, NO concentrations were close to zero as there were no significant NO emissions in the marine boundary layer. Water vapor concentrations showed the expected inverse correlation with the flight altitude with mixing ratios below the detection limit at high altitudes. As already suggested by the vertical $NO_2$ concentration profiles in Figure 4a, $NO_2$ concentrations obtained from CLD measurements were lower than $NO_2$ concentrations from miniDOAS measurements and PSS calculations (and sometimes even below zero) at high altitudes and

higher at low altitudes. Please note that the CLD $NO_2$ data shown in Figure 5 were processed as described earlier. This includes the interpolation of the background and calibration measurements which were performed 6 times for MF10 and 4 times for MF12. A potential variation of the background or the sensitivity of the channels between two CB-cycles would therefore be unaccounted for. We show the background measurements in the top data trace of each subfigure in Figure 5 by red dots and the interpolation as red dashed line. For MF10, the measured background in the $NO_c$ channel varied between 85 and 110 pptv and

for MF12 between 87 and 109 pptv. Calculated PSS $NO_2$ concentrations ranged between 1 and 93 pptv for MF10 and between 2 and 46 pptv for MF12. Local maxima mainly accompanied peaks of nitric oxide which is a result of the NO dependence of Equation (5). $NO_2$ concentrations measured by the miniDOAS instrument varied between 2 pptv and 39 pptv for MF10. For MF12, concentrations were $15 \pm 16$ pptv and generally lower for low altitudes and higher for high altitudes. $NO_2$ concentrations measured by the CLD instrument ranged from -224 to 317 pptv for MF10 and from -153 to 384 pptv for MF12. It is striking

that the maxima were obtained simultaneously with a sharp decrease in altitude accompanied by an increase in water vapor





(a) MF10

(b) MF12

**Figure 5.** Temporal development of NO, water vapor, and calculated and measured $NO_2$ exemplarily for measurement flights 10 and 12.

concentration. For each measurement flight here shown, this phenomenon was observed twice, indicated by the orange dashed lines. For example, the research aircraft descended from 12.6 km to 3.9 km at 13:30 UTC during MF12. At the same time, the CLD $NO_2$ concentration increased from an average of $23 \pm 14$ pptv between 13:00 and 13:30 UTC to its maximum of 384 pptv

at 13:45 UTC when reaching the new lower flight altitude. Water vapor measurements were incomplete before 13:00 UTC, but it can be assumed that they were constant and close to zero at 12.6 km altitude, rising to $9.5 \pm 0.7 \times 10^3$ ppmv on average after reaching 3.9 km (+ 15 minutes). Similar observations were made for MF12 at 18:30 UTC and for MF10 at around 13:30 and 18:00 UTC, in each case accompanied by a decrease in altitude and an increase in water vapor concentrations. The observed $NO_2$ peaks appeared only for the CLD measurement, and not for the values from PSS calculation or miniDOAS measurement which underlines the instrumental cause. The time series for the measurement flights MF13, MF14 and MF15 shows similar results and can be found in Figure S3 of the Supplement.

We hypothesize that these observations were influenced by a surface effect in the blue light converter which has a highly porous inner surface as described earlier. This material can adsorb atmospheric compounds, such as NO, and desorb them at a later stage (for example supported by an increase in humidity), which we will refer to as memory effect in the following. In a series of laboratory studies, we have investigated the impact of NO concentrations and humidity on the effects described above and particularly in regard to the instrumental background.

## 3.2 Laboratory experiments and implications for CAFE Africa

We propose that the memory effect described in the previous section is strongly affected by NO molecules and is dependent on changes in the introduced relative humidity. In order to show this, we have conducted different experiments in the laboratory to investigate the instrumental background produced by the photolytic converter in the $NO_c$ channel. Beside NO and $H_2O$, we suggest that one or more additional factors affect the observed background signal which are connected to the light of the LEDs and which we discuss at the end of this section.

For the first set of experiments, we exposed the converter (LEDs switched-on) to 16 ppbv of NO for 2 h followed by 4 h zero air measurements. The first experiment was carried out under dry conditions, sampling NO and zero air directly from the gas cylinder. For the second experiment, we introduced water vapor by passing zero air through a washing bottle with deionized water before entering the instrument. The thus obtained relative humidity was $\sim 95 \%$ at ambient temperature and decreased over time with decreasing water temperature (through evaporation). For the third and the fourth experiment, zero air was humidified only for the zero air measurement and the NO measurement, respectively. We repeated the latter introducing a lower relative humidity of $\sim 35 \%$ and obtained the same result.

Figure 6 shows the temporal development of the background signal in ppbv in the $NO_c$ channel after 2 h NO measurement with a concentration of 16 ppbv. The red line shows the experiment under dry conditions and the blue line presents the experiment under humid conditions. For the yellow line, we performed the NO measurement under dry conditions and the following background measurement under humid conditions. The green line shows humid NO measurement and dry background measurement. Figure 6a presents the results obtained with the conventional blue light converter modified with a quartz glass cylinder. All experiments showed a decreasing background signal over time in the $NO_c$ channel. For comparison, the background signal in the NO channel did not show a trend over time for any experiment which is presented in Figure S4 of the Supplement. This indicates that the converter adsorbs NO molecules, for example during NO measurements, and desorbs them during zero air measurements. At t = 0 min, the background signals for all experiments were approximately the same between 90 and 100 pptv,





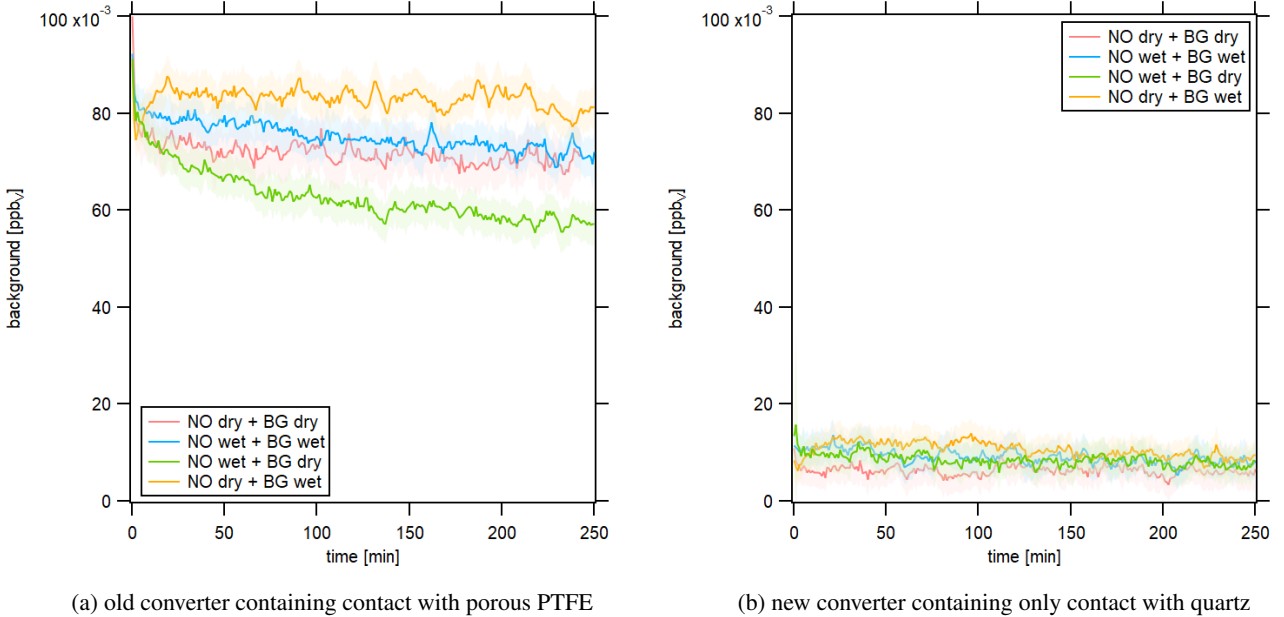

(a) old converter containing contact with porous PTFE    (b) new converter containing only contact with quartz

**Figure 6.** Instrumental background in response to dry and humid conditions after 2 h of NO calibration at 16 ppbv in the $NO_c$ channel. The LEDs of the photolytic converter were switched on.

but showed a different development over time. The strongest decline by around 25 % over 250 minutes was observed for dry background measurement after humid NO measurement (green curve). For the opposite case, humid background measurement

after dry NO measurement, the signal decreased by around 5 % over the observed time period (yellow curve). Performing the whole experiment under dry or humid conditions (red and blue curve, respectively) had a similar outcome with a background signal decrease of approximately 10 %. These observations indicate that the background measurements in the $NO_c$ channel which were performed during CAFE Africa and used for the data processing according to Equation (4) were consistently too high as they were only run for two times one minute (per CB-cycle) and therefore did not represent the actual instrumental

background but an artifact signal. In the laboratory, we observed the strongest effect for a dry background measurement following a humid NO measurement which was a likely scenario for the ambient monitoring during CAFE Africa as zero air for a background measurement was sampled from a gas cylinder (dry conditions) and the measured ambient concentration was subject to ambient meteorological conditions. This would explain the occurrence of negative $NO_2$ concentrations obtained from CLD measurements as mentioned earlier. These experiments also suggest that water molecules might not just promote

adsorptive or desorptive processes, but participate in the surface allocation themselves and compete with NO. Following this hypothesis, the surface spaces would fill with NO during an NO measurement under dry conditions and with both, NO and $H_2O$ molecules, under wet conditions. A subsequent background measurement under dry conditions would then lead to NO (and possibly $H_2O$) desorption. For a subsequent background measurement under humid conditions, $H_2O$ molecules could actively replace NO molecules because of their higher surface affinity, leading to a temporally longer "leakage" of NO and a



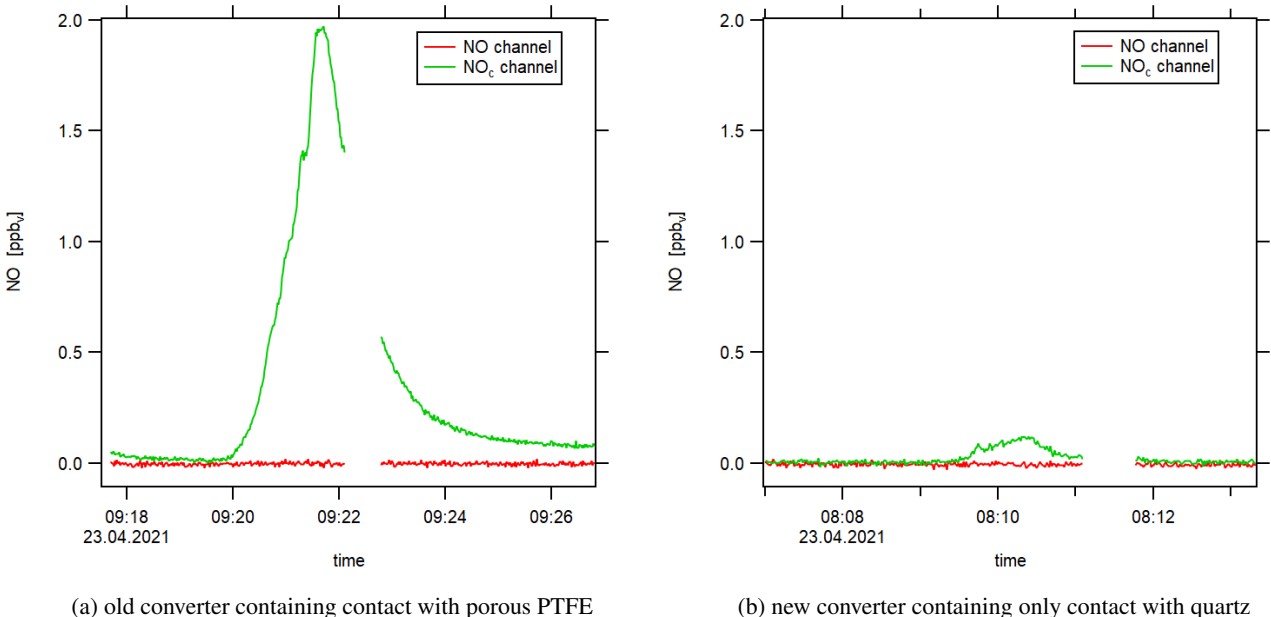

(a) old converter containing contact with porous PTFE

(b) new converter containing only contact with quartz

**Figure 7.** Instrumental background (zero air measurement) when heating the switched off photolytic converter with a heat gun. The small data gaps are due to prechamber measurements.

slower decrease in the background signal. The described effect would be highest for a dry NO measurement which gives the maximum NO surface coverage and a wet zero air measurement (yellow), while it would be lowest for the opposite case for which we observed the lowest background signal after 250 minutes (green). This hypothesis fits well with the observations during CAFE Africa presented in Section 3.1.3 where we observed a sharp increase in the $NO_2$ signal along with rapid increases in water vapor concentration. $H_2O$ molecules could replace adsorbed NO molecules (or other atmospheric compounds) which
were detected by the CLD as an artifact signal. Because of the conversion efficiency of 24.4 % of the blue light converter, the signal difference from the NO and the $NO_c$ channel was multiplied by a factor of around four (compare Eq. (4)). Therefore, the resulting $NO_2$ signal was distorted by four times of the actual desorbed NO explaining the large peaks accompanying altitude descents.

We repeated the same laboratory experiments using the new photolytic converter. The resulting temporal development of
the instrumental background is presented in Figure 6b which shows significant improvements compared to the conventional converter. The background in the $NO_c$ channel was many times smaller for the new converter with mixing ratios of around 10 to 15 pptv and was, most importantly, constant over time. For stationary long-term experiments it could be possible to measure the instrumental background on the scale of hours. However for aircraft measurements and the accompanying rapid air mass changes due to the high aircraft velocity, it is vital to obtain a reliable background measurement within a short
time interval, which would be possible with the new converter. Furthermore, changes in humidity did not seem to impact the measurement as all four experiments show the same result. This too, suggests the suitability of the new converter in



aircraft measurements or generally field studies which are impacted by high and changing humidities. Performing zero air measurements after $NO_2$ measurements had a similar outcome for each of the applied converters. The background measurement in the blue light converter showed a decreasing trend over time while it was constant and significantly smaller in the new photolytic converter.

Our assumption that the observed effect is associated with NO molecules, not $NO_2$ molecules or other $NO_x$ containing trace gases, is supported by an experiment where we heated the blue light converter with switched-off LEDs with a heat gun and observed a sharp increase in the $NO_c$ channel during zero air measurement (following NO calibration measurement). The increase in temperature promoted the desorptive process and had to include NO molecules. Otherwise, the CLD would not have detected any increase in the signal as the converters' LEDs were switched-off and $NO_2$ could not form NO via the photolytic reaction. We show the result of the heating experiment in Figure 7a. The converter surface was heated for two minutes (under constant movement of the heat gun) at a distance of around 10 cm during zero air measurement. We estimate the surface temperature to not have exceeded 200 °C. We observed a peak NO concentration of 2 ppbv ($NO_c$ background was 0 ppbv). In comparison, Figure 7b shows the experiment repeated with the new quartz glass converter which showed a small increase in the $NO_c$ signal, too, but approximately one magnitude smaller compared to the conventional converter. The qualitative outcome of this experiment was the same with the LEDs switched-on as well as with preceding $NO_2$ (instead of NO) measurement. Please note that a direct comparison of experiments regarding adsorptive and desorptive processes with switched-on and -off LEDs is difficult because the operation of the LEDs increases the temperature within the converter which - as shown above - strongly impacts the surface allocation.

Beyond that, we performed an experiment to investigate how NO calibration measurements affect subsequent zero air measurements in response to different NO concentrations. Figure 8 shows the influence of the preceding NO concentration level on the following first 5 minutes of zero air measurement. We have performed 30 minute NO calibrations with NO concentrations between 0.25 ppbv and 10 ppbv. Red data points represent the background of the NO channel and green data points show the background of the $NO_c$ channel. Background concentrations in the NO channel were independent of preceding NO concentrations. In contrast for the conventional converter, background concentrations in the $NO_c$ channel increased with increasing NO concentrations and leveled off for high values as shown in Figure 8a. Measured NO concentrations during CAFE Africa were between 0 and 1 ppbv and were therefore situated in the rising part of the curve. That shows that background measurements during CAFE Africa were not only too high, but also depended on the preceding NO concentration. We tried to retrospectively correct the $NO_2$ data with a lower instrumental background as obtained from laboratory investigations after several hours of zero air sampling. However, it was not possible to quantify the effect of varying, preceding NO levels. Additionally, the impact of humidity had the exact opposite effect on the $NO_2$ measurements. While the higher than actual instrumental background led to lower than actual $NO_2$ concentrations, increases in humidity triggered higher than actual $NO_2$ concentrations. Figure 8b shows that the development of the background in the $NO_c$ channel did not depend on the preceding NO concentrations for the new photolytic converter. The instrumental background was $12 \pm 1$ pptv and constant over the whole experiment. The detected background signal for the $NO_c$ channel disappeared when switching off the light of the UV-LEDs. A possible explanation can be a trace concentration of $NO_2$ in the utilized synthetic air as mentioned earlier.


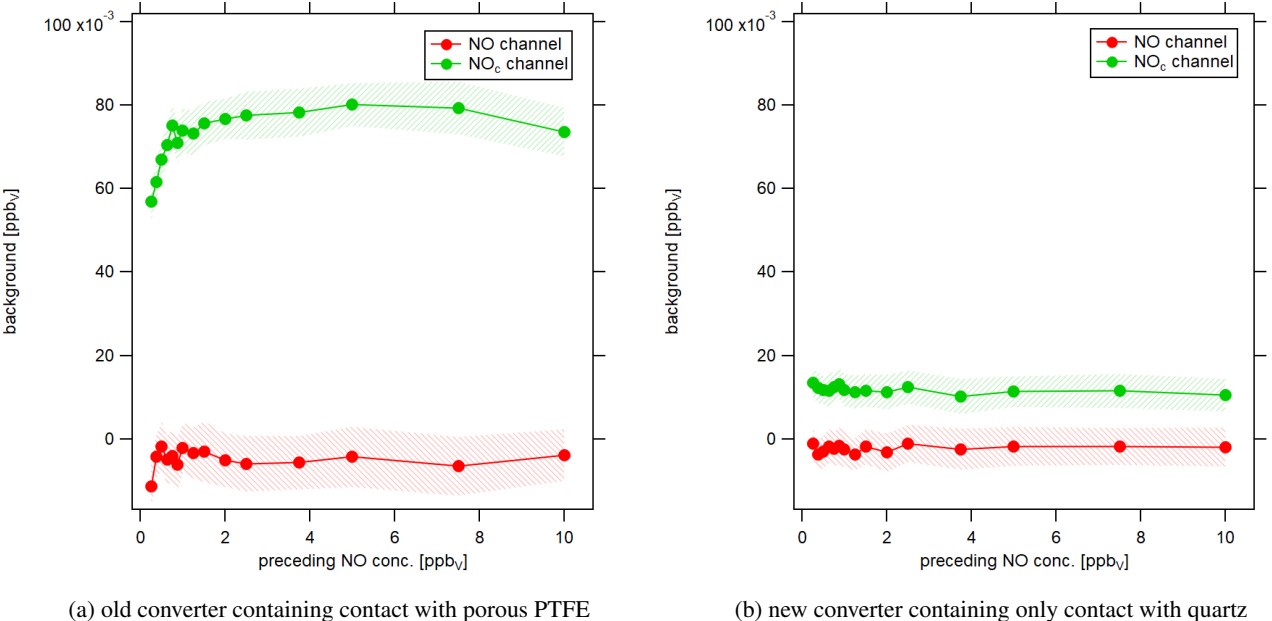

(a) old converter containing contact with porous PTFE     (b) new converter containing only contact with quartz

**Figure 8.** Instrumental background in response to different preceding NO concentrations under dry conditions. Each data point shows the level of the signal of the first 5 minutes of background measurements after NO calibrations.

While we have shown above that NO and humidity strongly affect the background measurements in the conventional blue light converter, it is likely that there are other factors contributing to the observed effects, too. When switching off the LEDs in the conventional converter, the observed instrumental background decreased rapidly (too rapidly for a sole temperature effect)
which we present in Figure S5 of the Supplement. This suggests that the light of the LEDs impacts the instrumental background in the $NO_c$ channel. Many other compounds can be photolyzed to form NO, such as PAN, $ClNO_2$ or $BrONO_2$. However, their absorption cross sections suggest no interference at 398 nm, the spectral output maximum of the LEDs (Reed et al., 2016; Pollack et al., 2010). Only small interference could occur with HONO and $NO_3$ at the edge of the spectral output and this would require the presence of these compounds in the converter which should not be the case for the described laboratory
investigations, but is conceivable given the memory effect observations. We have performed an uptake experiment for $HNO_3$ (nitric acid) to investigate the adsorptive capacity of the converters. $HNO_3$ in zero air (2500 sccm) was first routed through a bypass and after reaching a constant signal, the gas flow was changed to include the converter. The $HNO_3$ concentration behind the converter was monitored via chemical ionization mass spectrometry (CIMS). Figure S6a and S6b of the Supplement show the resulting adsorption behavior for the old and the new converter, respectively. When directing the gas flow through the old
converter, the detected $HNO_3$ flux decreased rapidly by around a factor of four and we did not observe the signal return to its initial value within 40 minutes (which is when we terminated the experiment). This indicates a high absorptive capacity or a decay of $HNO_3$ in the converter (or both). Integration of the $HNO_3$ flux shows that the converter took up approximately $1.7 \times 10^{16}$ $HNO_3$ molecules in the considered time frame. In contrast for the new converter, the $HNO_3$ flux decreased, too, but



returned to its initial value within 10 minutes while it adsorbed only $\sim 1.7 \times 10^{15}$ HNO$_3$ molecules. The observed adsorption

capacity can be minimized by coating the quartz surface with FEP (fluorinated ethylene propylene) which provides a highly

hydrophobic surface (Neuman et al., 1999; Liebmann et al., 2017). The number of adsorbed HNO$_3$ molecules was by a factor of

4 - 5 smaller compared to the non-coated quartz converter (Figure S6c). We did not observe any differences between the coated

and the non-coated quartz converter regarding the experiments investigating the role of NO and humidity presented above.

This uptake experiment shows the high adsorptive capacity of the old converter in comparison to the new quartz converter. In

the case of HNO$_3$, which could have been adsorbed by the converter e.g. during stratospheric measurements, we hypothesize

a potential source of NO or NO$_2$ through for example surface-catalyzed chemistry, possibly involving the light of the LEDs or

elevated temperature. Again, this experiment underlines the superiority of the new photolytic converter over the conventional

blue light converter and suggests the applicability for ambient and airborne measurements.

## 4   Conclusions

In this study, we have investigated a modified conventional blue light converter with a highly reflective and porous inner

surface made from optical PTFE regarding its application during the research aircraft campaign CAFE Africa which took

place in August and September 2018 around Cabo Verde. We have identified a memory effect in the blue light converter

which is affected by humidity, especially by rapid changes in water vapor concentrations, as well as preceding NO levels,

which is particularly relevant for the low NO$_2$/NO ratio in the upper troposphere. The high adsorptive capacity regarding other

atmospheric trace gases such as HNO$_3$ and the light of the LEDs could additionally play a role in the observed effects. Because

of the complex correlations between these parameters it is not possible to retrospectively correct the NO$_2$ signal measured

during CAFE Africa to receive reliable data. Instead, we suggest the application of a new photolytic converter made from

quartz glass in order to prevent the gas flow from contact with the porous PTFE surface which can additionally be coated with

FEP to obtain highly hydrophobic properties. Laboratory results indicate a high suitability of the newly developed converter

in aircraft measurements which - looking into the future - should be investigated in detail in order to improve in-field NO$_2$

measurements. With an improved instrumental background, other important questions of current atmospheric NO$_2$ research

such as deviations from photostationary state NO$_2$ in remote locations or interferences from NO$_y$ species could be addressed

and investigated more easily.

*Data availability.*   Data measured during the flight campaign CAFE Africa are available to all scientists agreeing to the CAFE Africa data

protocol.

*Author contributions.*   HF had the idea. CMN and HF designed the study. CMN analyzed the data and wrote the manuscript. CMN performed

the laboratory experiments. CMN, UP and HF designed the new photolytic converter. IT provided CLD NO$_x$, CO and CH$_4$ data for CAFE

Africa. Photolysis frequencies were received from BB. DM, MM, RR and HH measured and provided the OH and HO$_2$ data. KP ad FK





mesaured and provided the $NO_2$ miniDOAS data. FO measured and provided the $O_3$ data. $H_2O$ data were received from MZ. RD and JNC
measured and provided the PAN data. HF, JL and HH had a large contribution in the operation and planning of the aircraft campaign.

*Competing interests.*  Hartwig Harder is a member of the editorial board of the journal.

*Acknowledgements.*  We acknowledge the collaboration with the DLR (German Aerospace Center) during CAFE Africa. This work was
supported by the Max Planck Graduate Center with the Johannes Gutenberg-Universität Mainz (MPGC). FK and KP acknowledge the
support given by the Deutsche Forschungsgemeinschaft (DFG) through the projects PF 384-16, PF 384-17, and PF 384-19.

*Financial support.*  The article processing charges for this open access publication were covered by the Max Planck Society.



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
