# Peer review of "Modification of a conventional photolytic converter for improving aircraft measurements of NO2 via chemiluminescence."

_Atmospheric Measurement Techniques, 2021_

## Author Comment (AC1)

**Referee 2:**

*The measurement of NOx in remote air is very challenging, in particular because of the difficulty of accurately determining the NO2 artefact of photolytic convertor-CLD (P-CL) measurements, the current gold standard technique for accurate NOx measurements.*

*This manuscript, while not entirely novel as the authors point out in terms of presenting an alternative quartz glass converter for P-CL measurement of NO2, is very useful especially because of the discussion of laboratory experiments to investigate the instrumental background produced by the photolytic converter in the NOc channel and the characterisation of an improved convertor.*

*I recommend publication after the following points have been addressed:*

We would like to thank the referee for the positive feedback and the recommendation for publication.

*Pg 5. A $NO_2$-> NO conversion efficiency of 14% (or even 20% for the original convertor) is very low (i.e. Andersen et al. 2020 report CEs of >50% ). I suggest the authors mention that a higher CE is desirable for improved accuracy and perhaps suggest ways this could be implemented.*

We agree with the referee that a higher CE would be desirable. This could easily be implemented when operating the converter at higher pressures. However, for aircraft measurements, an additional uncertainty would result from calculating altitude-dependent CEs. Additionally, operation at higher pressure also increases the fractional dissociation of thermally unstable NOx reservoir species which decay in the converter which is shown in Figure S1a of the Supplement. The referee made a good point here and this is a topic which we want to investigate in more detail in the future. We have added some text for clarification.

Lines 152 ff.: Please note that the low conversion efficiencies in both converters result from the operation at low pressures which we have implemented to pursue aircraft measurements where altitude changes are accompanied by pressure variations. Operating the converter at lower than minimum ambient pressure levels (max. ~15km flight altitude) has the benefit of a constant conversion efficiency. The fractional dissociation of thermally unstable NOx reservoir species increases with increasing pressure in the converter which can be seen in Figure S1a of the Supplement. On the other hand, a higher conversion efficiency would be desirable for improved accuracy of the measurement.

*Pg 5. "Therefore, a pre-chamber measurement is operated for 20 seconds every 5 minutes where ozone is added to the sample gas flow". What is the efficiency of the pre-chamber volume (i.e. how much of the added NO from the calibration gas reacts with O3) ? It should be >98% or so.*

For the NO channel, the efficiency of the pre-chamber is >96% and for the NOc channel it is close to 100%. We have added this information to the main text.

Lines 166 ff: The residence time in the pre-chamber allows for the reaction of O3 and NO and the relaxation of NO2* before entering the main reaction chamber (pre-chamber efficiency > 96% for the NO channel and ~ 100% for the NOc channel)

*Pg 5 Ln 141. The "constant temperature of 25oC" in the convertors is not monitored, and so could presumably be a lot higher when the LED lights are on. The authors rightly point out that accurate determination of this temperature is critical for the calculations of the NO₂ artefact. It would also be highly beneficial to perform measurements of e.g. PAN degradation to confirm the artifact calculations (and, indirectly, indicate the temperature in the chamber).*

The "constant temperature of 25°C" in this sentence refers to the reaction chambers where the reaction of NO and O3 takes place, not the converter where NO2 is converted to NO. The temperature in the reaction chambers is constantly monitored and has a maximum variation of ± 0.1°C.

The temperature in the photolytic converter was not monitored during CAFE Africa. However, the temperature of the gas outflow was measured to be 40°C which we assume to be identical to the inner temperature for our calculations. The temperature in the alternative quartz converter is likely lower as the sample gas does not get in direct contact with the LEDs.

Based on the temperature of the conventional converter and the short residence time we calculate that only 0.1 % of any PAN would thermally decompose to NO2. Therefore, a PAN mixing ratio of 200 pptv would result in release of only 0.2 pptv NO2. This is consistent with findings by Reed et al. (2016).

Lines 159 ff.: Additionally, the sample gas flow in the type 2 quartz converter does not have contact with the LEDs which likely minimizes the sample gas heating and consequently the thermal interferences when passing through the converter.

Lines 199 f.: Please note that it was not possible to measure the temperature inside the converter. Instead, the temperature of the gas outflow from the converter in the ring channel was measured which we equate to the inner temperature.

*Ln 178. "Please note that the instrumental background for the NO data was determined by nighttime measurements of NO instead of zero air measurements ..." How often was night-time NO determined and what was the variability?*

The instrumental background for the NO data was determined from one night-time flight on August 26, 2018 and was 5.0 ± 5.3 pptv (1s integration time). In comparison, zero-air measurements throughout the campaign were on average 4 ± 7 pptv which is in close agreement. We have added text for clarification.

Lines 207 ff.: Please note that the instrumental background for the NO data was determined at 5.0 ± 5.3 pptv during a nighttime measurement on August 26, 2018 of NO as presented by Tadic et al. (2021) and previously described by Lee et al. (2009). The instrumental background determined via zero-air measurement was similar with 4 ± 7pptv (Tadic, 2021) (measured four to six times per MF).

*Page 7. An calculation of uncertainty for both NO and NO2 measurements is missing from the Experimental section.*

Thank you for noting this. We have determined the precision from the reproducibility of the NO calibrations and the detection limit from the reproducibility of the zero air measurements. This gives a precision of 3% (1σ). The accuracy of the secondary NO standard is 4%. For the NO channel and the NOc channel using the alternative quartz

converter, the detection limit is around 5pptv. When using the old converter, the detection limit in the NOc channel is more difficult to determine due to the observed memory effects and is estimated at >10pptv. We have added the information to the manuscript.

Lines 180 ff.: The precision is determined from the reproducibility of the NO calibrations and is 3% (1σ). The NO concentration is 4.96±0.21ppmv which gives a 4% uncertainty of the used secondary standard. The resulting NO calibration mixing ratio is 15.8±0.7ppbv. The detection limit is given by the reproducibility of the zero air measurements which is around 5pptv for the NO channel and the NOc channel using the type 2 quartz converter. The detection limit is higher when using the type 1 converter, but difficult to determine due to the observed memory effects and estimated at >10pptv.

*Ln 193 "Please note that these data (OH and HO2) are still preliminary"    Are final data yet available?  This would be highly desirable since HO$_2$ and OH are required for the calculation of [PNA], and CH$_3$O$_2$ is calculated from HO$_2$ and required to calculate MPN.*

We regret that final data are still not available. The high uncertainty of the data is based on difficulties associated with the calibration method for the data. We do not expect any changes to be significant.

*Figure 5.  Please include all data in the figure legend (including BG) and explain the orange dotted lines in the caption.  The word "exemplarily" is not needed in the caption.*

For a better overview of the figure we would prefer to limit the legend to the entries which are necessary for distinguishing the data. This is only relevant for the NO2 data traces while the other traces are specified by the according y axis label. BG is the abbreviation for background and the orange dashed lines indicate the simultaneous occurrence of an aircraft descent, an increase in water vapor concentrations and a peak in the NO2 CLD data. We have added text in the caption for clarification and removed the word "exemplarily".

Figure 5: Temporal development of the instrumental background (BG), NO, water vapor, and calculated and measured NO2 for measurement flights 10 and 12. The orange dashed lines indicate the simultaneous occurrence of a rapid decrease in altitude, an increase in water vapor concentration and a peak in the NO2 CLD data.

*Lns 352 onwards.  The authors demonstrate convincingly that memory effects of the porous convertor coupled to water vapour changes are a strong driver of changes in the instrumental NOc background.  However, the adsorbing/desorbing of NO molecules will likely also be affected by pressure as well.  Could the authors comment on this?*

The memory effect is related to processes which occur in the converter which is held at a constant pressure at all times and the observed processes can therefore be considered independently of pressure. Generally, variations in pressure will affect competitive adsorption of NO and other air molecules and the transport of trace gases to the surface.

*Ln 400 onwards.  I congratulate the authors on their much improved photolytic convertor and its apparent stability and insensitivity to varying humidity and lack of memory effects.  I would recommend also that experiments are conducted with varying pressure to evaluate pressure-dependence of the background.*

We thank the reviewer for this friendly comment and the suggestion. We are very interested in the role of pressure and are planning some studies on how pressure affects the conversion efficiency and whether it is more favorable to perform aircraft measurements with a constant, but low CE or a high, but variating CE.

*Ln 470 onwards. In the Conclusions section, the authors could consider adding recommendations on airborne NO2 measurements by P-CL, i.e. avoiding constant altitude changes in flight, which will inevitably change the background, and ensuring sufficient background measurements at each altitude change. This would be useful for the community.*

We have added this advice regarding the use of the conventional blue light converter to the conclusions section. Regarding the quartz converter, we are confident that the background is not subject to changes as a consequence of altitude changes based on our laboratory investigations. However, this still needs to be confirmed by field experiments.

Lines 518 ff.: If a conventional blue light converter is still in use, we would suggest to avoid constant altitude changes in aircraft applications. Instead, we highly recommend the application of an alternative photolytic converter made from quartz glass (…)

---

## Author Comment (AC2)

**Referee 3:**

*The authors describe an implementation of a photolytic NO2 converter demonstrating it's' use in airborne atmospheric research – specifically high altitude aircraft measurements. The selective photolysis of NO2 using a narrow band UV source illuminating a quartz cell, followed by detection of the resulting NO by chemiluminescence has been the reference method of NOx determination since the early 2000s'.*

*With some comments addressed the manuscript can make a valuable contribution to AMT.*

We thank the referee for the positive feedback and the time to review our manuscript.

*General comments:*

*The authors don't give a clear rational for modifying of the Droplet Measurement Technologies (also Air Quality Design, and now Teledyne API) Blue Light Converter in the way they have, especially given the sub-optimal results.*

Some laboratory studies were performed prior to the CAFE Africa campaign as part of a master thesis project and showed the mentioned surface effects. The modification of the Droplet Measurement Technologies BLC with a quartz tube was implemented to account for these effects. Due to a small time frame for the preparations at that point it was not possible to develop a new converter as alternative and undergo the full authorization /certification process for the implementation. Therefore, a quartz tube was inserted which was a compromise regarding the available time frame and the reduction of surface effects. The results of the NO2 measurements from the campaign show that the modification was by far not sufficient and post-campaign a converter entirely made from quartz was developed and tested. We have conducted the laboratory experiments with the modified conventional converter in order to enable comparison with the aircraft experiments. We have added text for clarification and added the names of the successor companies distributing the commercial blue light converter. We changed the description of the converter to type 1 for the modified blue light converter and type 2 for the alternative quartz converter for easier distinction.

Line 132 f.: We use a blue light converter (type 1) purchased from Droplet Measurement Technologies in 2005 (later Air Quality Design, now Teledyne API) (…)

Line 138 f.: Please note that this modification was made prior to the CAFE Africa research campaign within a limited time frame and did not have the desired outcome.

*Why reuse the low-powered 1 W, 395 nm, UV Hex, Norlux Corp. LEDs when much more powerful units are available – and in fact are used in more recent BLCs (see: https://doi.org/10.5194/amt-9-2483-2016)?*

Thank you for noting this. We have implemented the use of new LEDs which fit the modified converter design and added text regarding the specifications.

Lines 150 ff.: The applied LEDs were purchased from LED ENGIN (San Jose, California, USA) (High Efficiency VIOLET LED Emitter LZ1-10UB00-01U6, 2-2.2W, 395-400nm peak wavelength).

*What is the rational for moulding the PTFE around the quartz envelope? Were no alternatives tried? Similar aircraft implementations from NOAA, NCAR and FAAM, plus the paper cited by Andersen et al., 2021 use quartz cells wrapped in baking foil! Vapour deposition of optical silver has also been used in the past with no benefit over simply wrapping.*

*I realise in advance the answer to the two previous questions may be due to certification hurdles of HALO/DLR.*

We have performed an experiment with aluminum foil instead of PTFE and found a slightly (by a few %) lower conversion efficiency. At the same time, the PTFE envelope provides a stable housing for the sensitive quartz tube. We have added some text in the manuscript:

Lines 147 ff.: The PTFE material was found to provide a higher conversion efficiency in the converter compared to aluminum foil and additional provides a stable housing for the sensitive quartz tube.

*The authors should note that their PLC/BLC implementation has remarkably similar characteristics to a once commercially available unit also marketed by Droplet Measurement Technologies – a glass envelope, shrouded in PTFE, with arrays of Norlux UV-LEDs at either end. In this case the volume is ~ 115 cm3 which is the main difference. It is well described in the paper by Pollack et al., 2010 which the authors cite. These NO2 converters were previously operated by NCAR and FAAM on their aircraft, though are long since retired.*

We have added this information in the manuscript.

Lines 95 ff.: The use of quartz glass in a blue light converter was also reported by Pollack et al. (2010) who compared the commercially available converter BLC-A manufactured by Droplet Measurement Technologies to other photolytic converters.

*There is a marked drop in photolysis frequency between the two BLC modifications (0.66 to 0.46) using presumably the same LEDs. The authors should discuss why this is the case; is it the design, or aging of the LEDs, change in sample gas temperature etc.*

Thank you for noting this. We have added the information of the new LEDs which are likely responsible for the drop in photolysis frequency. We will investigate further improvements regarding the photolysis frequency and the conversion efficiency in more detail in the future.

Lines 150 ff.: The applied LEDs were purchased from LED ENGIN (San Jose, California, USA) (High Efficiency VIOLET LED Emitter LZ1-10UB00-01U6, 2-2.2W, 395-400nm peak wavelength).

*There is no description of the aircraft inlet from which the NOx instrument samples. Is that heated? What is the residence time to the instrument? What is the sample line/cabin temperature? When discussing the uncertainty of airborne measurements these must be taken into account.*

The NOx instrument sampled from the aircraft inlet via a bypass line of an approximate length of 1.5 - 2m and an inner diameter of 3/8 inches (1/2 tubing) (≈0.95cm) giving a volume of around 500 cm³ (±20%). The flow was around 80 SLM and the pressure in the

line depended on the aircraft altitude. At ground level pressure the residence time to the instrument was around 0.4s. At 12km altitude the residence time was only <0.1s. The aircraft inlet was not heated, the cabin/sample line temperature was approximately 25°C. We have added this information to the manuscript.

Lines 194 ff.: A bypass line provided the instruments with air from the aircraft inlet for which the residence time depended on the ambient pressure level (for high altitudes <0.1s). The sample line temperature was approximately 25°C.

*There is no schematic of the instrument given so I must assume it is identical to F2 given in Tadic et al., 2020. Several elements of the design shown there may skew measurements, especially during your discussion of the effects of humidity.*

*Firstly, the sample flows through mass flow controllers in the high pressure side of the inlet system which a) increases hold-up/lag, but also provides plenty of stainless steel surfaces to form layers of water on. Secondly, the NO and NOx channels appear to have different volumes due to there only being an NO2 converter on the NOx channel and no dead cell on the NO channel. Necessarily, there are different surface areas between the two, and different volumes, thus data must be offset between the two channels to compensate for the different residence times which themselves must either be very carefully measured or modelled. I doubt for instance that the true residence time if the BLC is 0.34 seconds – this is more likely 1 e-folding time. Lastly, and most crucially, the flow of humidified ozone is switched between reaction and pre-chamber of the CLD, with the sample (of varying humidity) constantly passing through the pre-chamber. This results in wildly fluctuating humidity within the pre-chamber. Many airborne CLDs follow the scheme in Pollack et al. 2010 whereby the humidified ozone constantly passes through the pre-chamber and the sample is switched – this also acts to decrease the response time of the instrument by removing dead volume.*

*Ultimately, I don't think flaws in the instrument design can account for the humidity effects described, though they should be considered.*

The referee is correct that the schematic of the instrument is identical to Figure 2 in Tadic et al. (2020). We have clarified this in the text.

Thank you for pointing out the possible effects of humidity that could result from the design of the instrument. Please note that the NO and the NOc channel are structurally identical apart from the photolytic converter and two short (~20cm) Teflon lines for the gas in and outlet. We do not observe any humidity effects in the NO channel and therefore we can exclude that stainless steel surfaces or the varying humidity in the pre-chamber account for the described effects. These components are equally present in both channels. The referee is correct about the different residence times in the two channels, however we only observe the humidity effects in the type 1 blue light converter with the porous surface, not in the type 2 quartz converter. The additional residence time using either converter is the same (0.33 and 0.34s). We therefore conclude that the different residence times cannot account for the observed humidity effects either. We have added a note on this to the manuscript.

Lines 124 ff.: All NOx measurements were performed using a modified two-channel chemiluminescence instrument originally purchased from ECO Physics, Dürnten, Switzerland (CLD 790 SR) as described by Tadic et al. (2020) (Figure 2 presets the instrument schematic) (…).

Lines 432 f.: As we do not observe these effects in the NO channel (and the two channels are structurally identical), we can exclude that any of the humidity effects are caused by components in the instrument other than the photolytic converter.

*The whole discussion on possible mechanisms for NO/H2O selective/competitive sorption is highly speculative and hard to follow at times.*

We regret that the referee finds the discussion on possible mechanisms hard to follow. We agree with the referee that we cannot finally prove the exact mechanism taking place in the photolytic converter which is why we use the word "hypothesis" in our discussion. We would like to offer our readers a possible explanation for our observations which is consistent with the results of the experiments we performed. For our work, knowing the exact mechanism is not imperative, but our observations and ideas might be helpful for other researchers who would like to solve similar questions.

*A typical test flight when commissioning a new NOx instrument is to fly whilst adding an amount of NO well above ambient to the inlet, performing profiles, orbits, in-cloud, boundary layer, and free troposphere runs – a system which is performing well will show no deviation throughout the entire flight envelope.*

We would like to thank the referee for this advice. We aim to conduct these experiments when using the quartz converter during our next aircraft campaign. We will then get the chance to test our quartz converter under ambient conditions.

*Specific comments:*

*Line 86: this was also the conclusion of Reed et al., 2016 which is cited.*

We have added the citation of Reed et al., 2016 to this sentence.

Line 90 f.: The correct adjustment of the conditions, preferably including low pressure, high flow rates and small temperature variations, can minimize interferences which was also concluded by Reed et al. (2016).

*Line 117: 'commercially available' – all four example of the CLD 790 SR were built for DLR on special order, no?*

We purchased our instrument from ECO Physics around 2000. The reviewer is correct that four examples (as far as we know) exist. We have rephrased this sentence.

Lines 124 ff.: All NOx measurements were performed using a modified two-channel chemiluminescence instrument originally purchased from ECO Physics, Dürnten, Switzerland (CLD 790 SR) as described by Tadic et al. (2020) (...).

*Line 123: State the reason why the photolysis cell is operated at 110 mb i.e. this is a pressure height of ~50kft which is the service ceiling of HALO/G550.*

We operate the converter at a pressure level lower than the minimum ambient pressure so that the conversion efficiency from NO2 to NO is not dependent on the aircraft altitude, but a constant value in order to prevent an additional uncertainty from calculating altitude-dependent CEs. We have added some text for clarification.

Lines 152 ff.: Please note that the low conversion efficiencies in both converters result from the operation at low pressures which we have implemented in regard to aircraft measurements where altitude changes are accompanied by pressure variations. Operating the converter at lower than minimum ambient pressure levels (max. ~15km flight altitude) has the benefit of a constant conversion efficiency. The fractional dissociation of thermally unstable NOx reservoir species increases with increasing pressure in the converter which can be seen in Figure S1a of the Supplement. On the other hand, a higher conversion efficiency would be desirable for improved accuracy of the measurement.

*Line 124: The stated wavelength is 398 nm of the UV LEDs – the design wavelength is 395 nm – is this a typo or was it measured (and not shown/described)? If the latter then a lot of energy is being wasted outside of the quantum yield of NO2 which drops rapidly at ~400 nm.*

We have characterized the UV LEDs in the laboratory and the spectral emission showed a maximum at 397 nm with a full width at half maximum of 14 nm. We have corrected this in the main text.

Lines 132 ff.: We use a blue light converter (type 1) purchased from Droplet Measurement Technologies in 2005 (later Air Quality Design, now Teledyne API) equipped with UV-LEDs emitting at a wavelength of 397 nm (FWHM = 14 nm, as characterized in the laboratory) (…).

*Line 132: please state the j value along with the conversion efficiency i.e. 0.656 s-1*

We have added the j value in the main text.

Line 143: (j = 0.66 s-1)

*Line 138: please state the j value along with the conversion efficiency i.e. 0.457 s-1*

We have added the j value.

Line 150: (j = 0.46 s-1)

*Line 138: presumably the gas flow doesn't contact the LEDs either in the design depicted in F1b? This would in-turn lead to much less sample heating and have a large impact on any thermal artefacts.*

We agree with the referee. We have added this information to the manuscript.

Lines 159 ff.: Additionally, the sample gas flow in the type 2 quartz converter does not have contact with the LEDs which likely minimizes the sample gas heating and consequently the thermal interferences when passing through the converter.

*Line 178: limits of detection are only useful when an averaging time is stated, please add this, how many standard deviations are included in the determination of LOD and uncertainty? e.g. 5 pptv averaged over 10 seconds, 3 sigma uncertainty of 6% etc.*

Thank you for noting this. We have added the information.

Line 207: 5 pptv detection limit at 1 min integration time and 6 % relative uncertainty (1σ)

*Line 146: What is the residence time of the pre-chamber; same as the reaction chamber? What is the efficiency? What is the material?*

The pre-chamber volume is approximately 160 cm³ with a gas flow of 1.5 SLM and a pressure of ~100 hPa. This gives a residence time of approximately 0.6 s. Please note, that we do not measure the pre-chamber pressure and the residence time is therefore an estimate. The volume of the main chamber is approximately 380 cm³ and a pressure of 10 hPa which yields a residence time of around 0.15 s. The efficiency of the pre-chamber is >96% for the NO channel and ~100% for the NOc channel. The material of the chambers is gold-plated stainless steel. We have added this information to the manuscript.

Lines 166 ff.: The residence time in the pre-chamber allows for the reaction of O3 and NO and the relaxation of NO2* before entering the main reaction chamber (pre-chamber efficiency >96% for the NO channel and ~100% for the NOc channel).

Line 172 f.: The material of both the pre- and the main-chambers is gold-plated stainless steel.

*Line 282: '…monitoring system for pressure…' in an airborne system the pressure must always be known and/or controlled otherwise the conversion efficiency of NO2 to NO is unknowable, regardless of potential artefacts or not!*

We agree with the referee and have rephrased the sentence for clarification.

Lines 312 ff.: Based on these results, we recommend the implementation of a monitoring system for both temperature and pressure within the photolytic converter (…).

*Line 370: I'm not sure it is true that there are no trends in the NO channel signal – I think the scale may be helping here – perhaps fit the trends to remove any doubt or adjust the scale.*

[Figure]

We have added a linear fit to the measured background in the NO channel vs time for the two experiments and changed the scale for investigating the suggestion of the referee. There is no trend for most lines and if at all a downward tendency for the

measurement NO wet + BG dry which could be due to surface effects on other components such as metal surfaces within the instrument. However, the change is very small compared to the observations in the NOc channel and more importantly, the 3σ uncertainty is larger than (the 2σ uncertainty approx. equal) the observed decrease over 250 minutes and therefore not significant. We have added text in the manuscript to point that out.

Lines 403 ff.: For comparison, the instrumental background signal in the NO channel over time is presented in Figure S5 of the Supplement. The lines show no significant trend over time.

*Line 411 -414: the logic in this statement is flawed – you can only measure NO with a CLD, therefore you only saw NO in your experiment. You could do the same experiment with the BLC connected to a direct NO2 measurement, or a PAN-GC or a CIMS for that matter and would likely see may compounds desorb.*

We intended to say that NO has to be part of the observed effect in the blue light converter. We showed this by switching off the converter, so that only NO could be detected. The observed signal in the NOc channel increased when applying heat which we see as proof that NO is involved. However, other molecules can play a role, too, which we show exemplarily with the HNO3 uptake experiment (Figures in Supplement). We agree with the referee that our wording might be misleading and have rephrased the sentence.

Lines 447 ff.: Our assumption that the observed effect is - at least partly - associated with NO molecules is supported by an experiment where we heated the type 1 blue light converter with switched-off LEDs with a heat gun and observed a sharp increase in the NOc channel during zero air measurement (following NO calibration measurement).

*Sect 2.5: please define all the acronyms for the NOy species (MPN,PAN…) at their first use.*

The acronyms for the NOy species are first used and defined in the Introduction Section 1. However, we agree with the referee that it benefits the reading flow if we provide the acronyms in Sect 2.5, again and have adjusted the text accordingly.

Line 255: We consider the NO2 reservoir species PAN (peroxyacetyl nitrate), MPN (methyl peroxy nitrate) and PNA (pernitric acid).

---

## Author Comment (AC3)

**Referee 1:**

*This manuscript describes a photolytic converter for airborne measurements of NO₂. The focus of the paper is measurements of NO₂ in remote areas where mixing ratios are sub 1 ppb. The authors describe interferences and artefacts associated with a commercial photolytic converter that complicate these sub-1 ppb measurements and suggest modifications to the commercial converter that reduce the effects of artefacts. There is not an overabundance of publications about the nuances of NO₂ photolytic converters, and thus I believe this manuscript can be a value contribution to the atmospheric community.*

*I agree with the assessments and comments posted by the other referees. I had many of the same concerns. Thus, in an effort to streamline the review process, I have only included my additional thoughts here. I hope they are helpful.*

We would like to thank the referee for the feedback and the time to review our manuscript.

*Comments:*

*Title: I do not think that the word "new" is appropriate for the title. This is because there is not anything particularly novel about the modified converter. The use of a fully enclosed quartz cell with and without a reflective Teflon shroud is not new among the airborne research community. However, these PCL systems are typically custom built (e.g., Pollack et al., 2010, Jordan et al., 2020). I wonder if a better title could be "Modification of a commercial photolytic converter for improved aircraft measurements of NO₂ via chemiluminescence".*

We thank the referee for this suggestion and have implemented a new title for our manuscript.

Modification of a conventional photolytic converter for improving aircraft measurements of NO2 via chemiluminescence.

*Abstract: Please add another sentence or two to the abstract about the aircraft measurement findings related to the NO₂ reservoir species. This is first and foremost in the results section but seems to be lacking mention in the abstract. Also, the abstract is a bit misleading in that it highlights the memory effect as the key phenomenon. Yet, the observations from CAFÉ are likely a combination of phenomena that also include an artefact from the subtraction of two signal channels and a changing background.*

We agree with the referee and have added the missing information to the abstract. We would like to highlight that the observed effect which we refer to as memory effect associated with high preceding NO levels and changing humidities induces the observed artifacts and the changing background and summarizes the observed phenomena. More precisely, the instrumental background is increased through high preceding NO levels (almost linear correlation in low concentration NO range (< 0.5 ppb)). Rapid increases in humidity can additionally enhance the release of surface-adsorbed NO molecules and in combination with the low conversion efficiency (introducing a factor of around 5 to the NO2 concentration calculations via Equation (2)) produce the observed artifact signals.

Lines 6 ff.: We find the NO2 reservoir species MPN (methyl peroxy nitrate) to produce the only relevant thermal interference in the converter under the operating conditions

during CAFE Africa. We identify a memory effect within the conventional photolytic converter associated with high NO concentrations and rapidly changing water vapor concentrations, accompanying changes in altitude during aircraft measurements, which is due to the porous structure of the converter material. As a result, NO2 artifacts, which are amplified by low conversion efficiencies, and a varying instrumental background adversely affect the NO2 measurements.

*Line 6 and throughout: Maybe it is just me, but I find the use of the word "conventional" to be a bit bothersome. This is because the photolytic converters typically used aboard aircraft do not use the porous Teflon material with ring channel for gas introduction. The word "conventional" seems more appropriate for ground-based applications that utilize commercial monitors and commercial converters. It might help to clarify the difference in the text.*

Thank you for pointing this out. We have added some text for clarification and also now refer to the blue light converter as type 1 converter and to the alternative quartz converter as type 2 converter for avoiding confusion.

Lines 132 ff.: We use a blue light converter (type 1) purchased from Droplet Measurement Technologies in 2005 (later Air Quality Design, now Teledyne API) equipped with UV-LEDs emitting at a wavelength of 397 nm (FWHM = 14 nm) which is shown in Figure 1a. The converter was designed for airborne applications.

*Line 124: Can you add the year the BLC was purchased from DMT? This could help readers distinguish between the version of "conventional" BLC that you are using compared to other versions of "conventional" commercial BLCs.*

The BLC was purchased from DMT in 2005 while its patent was still pending. We have added the year to the text (see above).

*Line 134-142: I don't think the use of the words "new" or "newly-developed" are appropriate here since several existing converters already separate the sample flow from direct contact with the porous Teflon surfaces. Maybe a better word for the converter shown in Figure 1b is "modified" or "updated".*

We agree with the referee and now refer to the alternative quartz converter as type 2 converter throughout the text.

*Section 2.2: I understand the elimination of a night flight (MF11), but why were only MF10 and MF12 through MF15 used in this study? Were MF01 through MF09 not good candidates, was NO$_2$ data not collected during those flights, or was this phenomenon not observed during those flights?*

Unfortunately, NO2 data were not collected for MF01 – MF09 due to instrumental malfunction.

*Line 170: Can you add a figure (either here or in the SI) that shows the J-curve for your converters? The conversion efficiencies of the "conventional" BLC (20%) and your "updated" converter (14%) are very low. This is likely a function of your very low cell pressure, which when combined with the high flow rate, results in a short residence time in the photolysis cell. It would be helpful to see how each converter (the conventional versus the updated BLC) behaves over a range of residence times. Regardless, a note*

*should be included in the text to associate the low conversion efficiency with the low cell pressure, which is needed for high altitude measurements.*

We have added text explaining the low conversion efficiency and reasons for the low pressure operation of the photolytic converter. We have added the j-values for the two converters to the text.

Line 143: (j = 0.66 s-1)

Line 150: (j = 0.46 s-1)

Lines 152 ff.: Please note that the low conversion efficiencies in both converters result from the operation at low pressures which we have implemented to pursue aircraft measurements where altitude changes are accompanied by pressure variations. Operating the converter at lower than minimum ambient pressure levels (max. ~ 15km flight altitude) has the benefit of a constant conversion efficiency. On the other hand, a higher conversion efficiency would be desirable for improved accuracy of the measurement.

*Line 180: Is it reasonable to utilize a nighttime NO concentration instead of zero measurements for determining $c(NO)$ when the $c(NO_2)$ is determined from the subtraction of the NO measurement from $c(NOc)$ and $c(background_{NOc})$ determined from a zero? What is the magnitude of the difference between NO zeros at night versus NO zeros with an overflow of zero air? Has this difference been factored into an uncertainty calculation for $c(NO)$ and $c(NO_2)$? What is the overall measurement uncertainty for NO and $NO_2$? Also, what was the concentration and uncertainty of the NO standard used for calibrations. What was the effective calibration mixing ratio after dilution into the sample flow? It would be helpful to add these details to the manuscript.*

Thank you for noting the missing information which we have added to the text. We believe that the determination of the instrumental background in the NO channel via nighttime NO measurements is more reliable as there can always be traces of NOx in bottled zero air which additionally vary in concentration between different bottles. For the CAFE Africa field campaign, the instrumental background determined using nighttime measurements was 5 ± 5.3 pptv with a value of 4 ± 7 pptv determined using bottled zero air (for a 1 min integration time), so there was no significant deviation between the two methods. The overall 1σ measurement uncertainty for NO is 6% as described by Tadic et al. (2021). We do not state a measurement uncertainty for NO2 as it was not possible to appropriately evaluate the data due to the described memory effects. The data are not to be used for any scientific conclusions on NO2 concentrations in the upper troposphere and solely serve the demonstration of the problems associated with the conventional blue light converter. The concentration of the secondary NO standard used during the CAFE Africa campaign was 1.187 ± 0.036 ppmv. A flow of 8.6 sccm was diluted in 3.44 SLM of bottled zero air, giving a calibration mixing ratio of 2.97 ± 0.09 ppbv.

Lines 201 ff.: Zero air measurements and NO calibrations using a secondary NO standard (cylinder concentration of 1.187 ± 0.036 ppmv and calibration mixing ratio of 2.97 ± 0.09 ppbv) were performed regularly to determine the variability in the instrumental background and the sensitivity of the channels.

Lines 206 ff.: The NO data were processed as described by Tadic et al. (2021) (5 pptv detection limit at 1 min integration time and 6% relative uncertainty (1σ)). Please note that the instrumental background for the NO data was determined at 5.0 ± 5.3 pptv by a

nighttime measurement during measurement flight MF11 on August 26, 2018 of NO as presented by Tadic et al. (2021) and previously described by Lee et al. (2009). The background determined via zero-air measurement was similar with 4 ± 7 pptv (Tadic, 2021).

*Line 305: I wonder if the changes in background can be more carefully characterized in a future flight by overblowing the instrument inlet with zero air for the duration of a test flight (aka. a "null" flight). The in-flight instrument performance can be evaluated from changes in the background signal levels during vertical profiles and maneuvers, which can inform about precision, detection limit, motion sensitivity, and fluctuations with pressure and temperature. It can also inform about lags in the recovery of background signals with these perturbations. For high altitude chemiluminescence applications, it might also be interesting to characterize the PMT dark counts versus background in a future test flight by periodically turning off the reagent O3 injection.*

Thank you for this interesting thought. Performing a "null" flight could be interesting in regard to determining a stable background signal, which in theory should be constant and independent of ambient conditions. We believe we can achieve this stable background in the NOc channel with the newly implemented quartz converter and hope to pursue this idea in the future.

*Line 329: How do the $NO_2$ measurements change if you assume a constant background signal per altitude level? My first instinct would be that subtracting an interpolated background signal would contribute a good bit to the negative excursions in $NO_2$. Since the CLD 790 SR has two separate channels, is the BG trace in Figure 5 meant to be the background signal of the $NO_2$ channel? How does the background of the NO channel differ from that of the $NO_2$ channel with the LEDs on and off? Does the NO channel background also change with altitude or only the background measured through the converter? Can you add the NO channel BG as a trace in the Figure?*

We agree with the referee that in some cases negative NO2 values could result from interpolated background signals. However, we regularly observe negative NO2 values right before and after background measurements during constant flight levels e.g. MF10 between 10:30 and 12:30 or MF12 between 16:30 and 18:00 (BG signal almost constant). Therefore, the background generally being too high due to the described memory effect has a much larger impact on the occurrence of negative values as the interpolation. The background in Figure 5 is the background signal in the NOc channel – we have added this information in the Figure description. The background of the NO channel is usually very close to zero. When switching off the LEDs in the NOc channel, the signal approximates that in the NO channel which we present in Figure S6 of the Supplement. Please note that this experiment was only performed in the laboratory, but should be the same for in-field experiments. The background in the NO channel also showed small changed throughout each flight, but to a significantly smaller extent compared to the variations in the NOc channel:

[Figure]

For a better overview, we decided to show the background in the NO channel in the Supplement and refer to it in the main text.

Lines 361 ff.: In comparison, the background in the NO channel varied between 1 and 11 pptv for MF10 and between -2 and 6 pptv for MF12. We show the instrumental background for MF10 and MF12 in each channel in Figure S3 of the Supplement.

Figure 5: Temporal development of the instrumental background (BG) in the NOc channel, NO, water vapor, and calculated and measured NO2 for measurement flights 10 and 12. (…)

*Section 3.1.3: I admit, I found the logic of this section a little hard to follow. If I have this correct, the bulk of the discussion in this section is about the instances when NO2_CLD is enhanced yet there are no enhancements in NO2_PSS nor NO2_DOAS. The authors are claiming that the NO2_CLD enhancements are correlated with increases in water vapor as the aircraft descends. The authors associated the discrepancy to a hysteresis in the photolysis cell upon the introduction of water vapor. If this were the case, then I agree that a decrease in $NO_2$ back to baseline levels following the increase in water vapor with the lag time representing the memory effect time would be expected. However, the rising edge of the enhancement in $NO_2$ that starts to increase as a large step change in NO starts to decrease and that occurs earlier in time than the step change in water vapor concentration is not something that I would have expected from a memory effect phenomenon. This leads me to believe that the $NO_2$ peaks are more of an artefact of the NO channel subtraction, which is enhanced by a factor of 4 due to the correction for Ce, and less so from a memory effect of water vapor on the photolysis cell sampling surfaces.*

We respectfully disagree with the referee. The background in the NO channel and the NO measurements of the CLD are accurate and we can therefore be sure that the drop in NO concentration is correct and is also reasonable regarding the vertical atmospheric profile – presented in Figure 2a in Tadic et al. (2021). NO2 concentrations are determined via Equation (4) with the dominating uncertainties on c(NOc) and the instrumental background in the NOc channel. We show in Section 3.2 that the background is at all times too high and the subtraction therefore responsible for the observation of negative NO2 values. Unexpected NO2 peaks must be a consequence of difficulties in the NOc concentration which is most likely related to the memory effect in the photolytic converter. Both effects, negative data through higher than actual

background values and NO2 peaks through humidity induced desorption of (likely) NO, are enhanced by a factor of 4 (=1/Ce), as the referee points out correctly.

*From the manuscript (mainly the abstract, introduction, and conclusions), I am led to believe that the authors think the memory effect is the key phenomenon at play with the "conventional" converter. However, the results and discussion of the CAFÉ observations suggest that the signal subtraction, low Ce, fluctuations in background, and large changes in NO concentrations could also have been significant contributors to the observations. Thus, it seems a little misleading to only mention memory effects in the abstract and conclusions. It is my recommendation that the text in the abstract and conclusions be updated to reflect the observations and all possible factors that could have impacted the CAFÉ observations.*

As explained above, we consider the term "memory effect" to be the integral of the observed effects. The instrumental background fluctuations in the NOc channel and the problem of always being too high result from storage effects in the blue light converter. Higher preceding NO levels lead to a higher background in the measured concentration range because a short background measurement detects "leaking" NO molecules (and potentially other NOx containing compounds) from the porous converter material. The large NO2 peaks are likely a humidity triggered desorption of NO molecules (or trace gases that are converted to NO) when the aircraft is descending and therefore also related to the memory effects in the converter. The NO measurements in the NO channel yield reliable data and therefore atmospheric variations do not have adverse effects on the calculation of NO2 concentrations. We have added text in the conclusion for clarification.

Lines 513 ff.: More specifically, this includes the subtraction of a fluctuating, higher than actual instrumental background in the NOc channel yielding negative NO2 values as well as humidity triggered spontaneous desorption of stored molecules appearing as large NO2 peaks, both of which effects are amplified by the low conversion efficiency.

*Section 3.2: The UV artefact (Figure S5) of 0.1 ppb seems substantial for a sub-1ppb ambient measurement. How does the UV artefact factor into your subtraction calculations (e.g., eq. 4) and into the overall measurement uncertainty? What does Figure S5 look like for the updated converter?*

We agree with the referee that the UV artefact for the old converter is highly undesirable. When calculating the NO2 data, we assume that the artefact is equally present for background measurements, calibrations and ambient measurements and is therefore accounted for through the background subtraction. For the alternative quartz converter, the difference between the background with switched on and off LEDs is small and could potentially be due to a trace concentration of NO2 in the zero air gas cylinder which we describe in Lines 158 ff. We have added a Figure to the Supplement to show the effect for the alternative quartz converter and refer to it in the main text.

Lines 487 ff.: For comparison, Figure S6b shows that the effect of switching the LEDs on and off during zero air measurement is marginal when using the type 2 quartz converter.